

# Review Article: Permafrost Trapped Natural Gas in Svalbard, Norway

Authors: Thomas Birchall*[1, 2], Malte Jochmann[1, 3], Peter Betlem[1, 2], Kim Senger[1], Andrew Hodson[1], Snorre Olaussen[1]

[1]Department of Arctic Geology, The University Centre in Svalbard, P.O. Box 156, N-9171 Longyearbyen, Svalbard, Norway

[2]Department of Geosciences, University of Oslo, P.O. Box 1047, Blindern, 0316 Oslo, Norway

[3]Store Norske Spitsbergen Kulkompani AS, Vei 610 2, 9170 Longyearbyen, Svalbard. Norway

*Correspondence to: Thomas Birchall (Thomas.birchall@unis.no)

**Abstract.** Permafrost has become an increasingly important subject in the High Arctic archipelago of Svalbard. However, whilst the uppermost permafrost intervals have been well studied, the processes at its base and the impacts of the underlying geology have been largely overlooked. More than a century of coal, hydrocarbon and scientific drilling through the permafrost interval shows that accumulations of natural gas trapped at the base permafrost is common. They exist throughout Svalbard in several stratigraphic intervals and show both thermogenic and biogenic origins. These accumulations combined with the relatively young permafrost age indicate gas migration, driven by isostatic rebound, is presently ongoing throughout Svalbard. The accumulation sizes are uncertain, but one case demonstrably produced several million cubic metres of gas over eight years. Gas encountered in two boreholes on the island of Hopen appears to be situated in the gas hydrate stability zone and thusly extremely voluminous. While permafrost is demonstrably ice-saturated and acting as seal to gas in lowland areas, in the highlands it appears to be more complex, and often dry and permeable. Svalbard shares a similar geological and glacial history with much of the Circum-Arctic meaning that sub-permafrost gas accumulations are regionally common. With permafrost thawing in arctic regions, there is a risk that the impacts of releasing of sub-permafrost trapped methane is largely overlooked when assessing positive climatic feedback effects.

## Keywords

Permafrost; Top seal; Natural Gas; Cryosphere; Greenhouse Gas; Arctic; Greenhouse Gas; Hydrates.



## 1 Introduction

It is generally accepted that thawing permafrost results in the release of methane gas to the atmosphere (Knoblauch et al., 2018). Methane is a potent greenhouse gas and its release from permafrost acts as a positive climatic feedback loop (Boucher et al., 2009; Howarth et al., 2011; Lashof and Ahuja, 1990). The Arctic is particularly sensitive to climatic changes and Svalbard is even more so due to the West Spitsbergen Current (Divine and Dick, 2006; Van Pelt et al., 2016; Aagaard et al., 1987). Svalbard is, therefore, a critical site for studying the evolution of permafrost and sub-permafrost processes (Hornum et al., 2020; Hodson et al., 2019; Christiansen et al., 2010; Isaksen et al., 2000).

While methane emissions from thawing of the permafrost active layer is relatively well understood (Knoblauch et al., 2018; Vonk and Gustafsson, 2013), the prevalence and volumes of gas accumulations trapped beneath the permafrost "cryospheric cap" (Anthony et al., 2012) has been much less studied. Here we present evidence of such gas accumulations in Svalbard, where the relatively young permafrost (Gilbert et al., 2018) appears to be regionally sealing significant gas accumulations. The gas here may originate from biogenic or thermogenic processes (Hodson et al., 2019; Ohm et al., 2019) and may be in free-form or, under the right compositional and thermobaric conditions, in the form of natural gas hydrates (Sloan Jr et al., 2007; Betlem et al., 2019).

Occurrences of gas originating from within or below intervals of permafrost are typically identified in studies on natural gas hydrates and have been documented in both the Russian (Chuvilin et al., 2000; Makogon and Omelchenko, 2013; Yakushev and Chuvilin, 2000; Skorobogatov et al., 1998; Chuvilin et al., 2020) and North American Arctic (Bily and Dick, 1974; Collett et al., 2011; Kamath et al., 1987; Majorowicz and Hannigan, 2000; Nielsen et al., 2014).

Permafrost is defined as ground that remains at sub-zero (in Celsius) temperature for more than two consecutive years, regardless of fluid content. Physically speaking, ice-saturated permafrost possesses extremely good sealing properties (Keating et al., 2018). However, how effective it is as a top seal is uncertain, this is reflected in Svalbard by methane emissions at pingos (Hodson et al., 2019) where permafrost demonstrates its local sealing ability but also the prevalence of migration pathways through it. Abrupt changes in hydrogeological flow conditions at the base of permafrost also indicate the permeability-reducing nature of the permafrost interval (Hornum et al., 2020). In geological terms, permafrost is very short-lived which, in addition to being very shallow and potentially patchy, would typically preclude it from being regarded as a feasible seal for conventional hydrocarbon accumulations at geological time-scales of millions of years.

In Svalbard, methane migrating through near-coastal pingos shows characteristics of a biogenic origin (Hodson et al., 2019). Approximately three kilometres inland, analysis of gas encountered at the base of permafrost during drilling indicated a further contribution from thermogenic origins (Ohm et al., 2019). Several hydrocarbon source rocks are encountered in Svalbard, so traces of thermogenic gas are not particularly surprising. What is surprising, and the focus of this study, is the widespread occurrence, both spatially and stratigraphically, of gas accumulations at the base of permafrost in Svalbard. In this contribution, we therefore provide previously unpublished data from 41 boreholes to provide a systematic review of the occurrence of sub-permafrost gas



accumulations from Svalbard. We also analyse data from these boreholes to attempt to characterise the
permafrost, its thickness and sealing properties.

## 2 Geological and physiographic setting

The Svalbard archipelago is situated in the high arctic between 74° to 81°N and 15° to 35°E with sub-zero
average temperatures for eight months of the year. Despite this, due to repeated glaciations and the warming
effects of the West Spitsbergen Current, permafrost in Svalbard is not as thick as some other pan-arctic regions
(Humlum, 2005).
Permafrost in Svalbard ranges in thickness from more than 500 m in mountainous areas inland and thins to less
than 100 m near the coastlines (Humlum, 2005). Continuous sub-sea permafrost has not been shown to exist
offshore on Spitsbergen's west coast (Christiansen et al., 2010) and is not believed to be present offshore
elsewhere around Svalbard (Humlum et al., 2003; Landvik et al., 1988), although these areas have been little
studied in this respect and may feature locally discontinuous permafrost. Because of the West Spitsbergen
Current, temperatures are much warmer on the west coast than the east. Although poorly studied in eastern parts,
one can reasonably anticipate thicker permafrost due to colder temperatures, as is also shown by numerical
modelling of the permafrost-associated gas hydrate stability zone (Betlem et al., 2019). However, thicker
insulating snow coverage in coastal settings can also help in preventing winter heat loss from the ground and
limit permafrost growth (Humlum et al., 2003). In a more local context, permafrost in Adventdalen has been
relatively well studied with near-zero thickness on the coast rapidly thickening to approximately 150 m thick 3
km inland at the Longyearbyen $CO_2$ site and approximately 220 m thick in the valley at Janssonhaugen, some
fifteen kilometres from the coast (Isaksen et al., 2000; Harada and Yoshikawa, 1996; Gilbert et al., 2018).
The permafrost history of Svalbard is something of a contentious issue but it is important to understand as it
provides clues to the timing and rate of gas accumulation at its base. The driver for the permafrost evolution in
Svalbard is dependent on glacial settings rather than temperature changes. During the Weichselian glacial stage
(115 kya to 11.7 kya) Svalbard was covered by thick glacial ice, although the extent and thickness of this ice
cover is still debated (e.g. Gataullin et al., 2001; Lambeck, 1996: Winsborrow et al., 2010). Glacial striations in
several locations suggest that these glaciers were warm-based for at least parts of the Weichselian glaciation
(Humlum, 2005; Humlum et al., 2003). The frictional heat generated from the sliding of warm-based glaciers
likely thawed permafrost in major valleys (Humlum et al., 2003). Sedimentological and cryostratigraphic
analysis of boreholes in Adventdalen support this (Gilbert et al., 2018), suggesting permafrost here has formed in
the past few thousand years following the dynamic retreat of these warm-based glaciers and ice streams. Whether
permafrost survived the Weichselian glaciations is dependent on the persistence of ice-free zones and/or cold
based glaciers. Because of this, permafrost in highland areas was more likely to have survived, possibly for
several hundred thousand years, through multiple glacial events (Humlum et al., 2003). There is also strong
evidence of lowland areas in north-western Svalbard being ice-free at this time which may have enabled the
persistence of much older permafrost (Landvik et al., 2003). Valley settings are pertinent to this study as the
majority of wellbores have been drilled in valleys or near to the coast for logistical reasons.



Permafrost often poses a challenge to geologists, particularly for drilling boreholes (Vrielink et al., 2008), and
acquiring and processing seismic data (Schmitt et al., 2005; Johansen et al., 2003). This is because it changes the
properties of shallow unlithified sediments to become much more rigid and cemented by ice. Therefore, the
permafrost interval has much faster seismic velocities and can lose mechanical competence as it is drilled
through with heated or saline fluids. The near-surface rocks in Svalbard are typically well cemented and very
rigid due to deep burial and subsequent uplift.

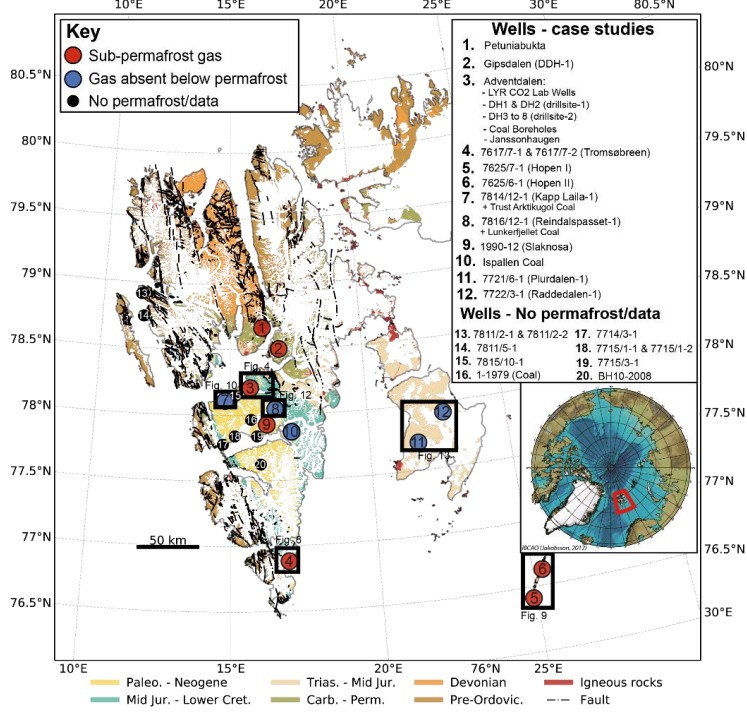


**Figure 1 – Map of Svalbard with boreholes and areas of interest investigated in this study.  Geological data is courtesy**
**of Norwegian Polar Institute (Dallmann et al, 2015). The locations of maps shown in later figures are highlighted.**
In a tectonic context, Svalbard represents the exposed north-western part of the Norwegian Barents Sea
continental shelf.  Other than the upper Cretaceous and parts of the Neogene, Svalbard exhibits a continuous
stratigraphic record from the Devonian to present (Steel and Worsley, 1984). Figure 1 shows the distribution and
ages of outcrops in Svalbard and the key wellbore sites for this study. Palaeozoic events from the Caledonian
(Gasser, 2014) and Ellesmerian-Svalbardian (Piepjohn, 2000) orogenies are predominantly recorded in the
remote northern parts of Svalbard. From the Late Carboniferous to Permian, mixed shallow marine rocks were
deposited in local basins (Smyrak-Sikora et al., 2019; Bælum and Braathen, 2012). From the Triassic to Early
Cretaceous, clastic deposition occurred in regional-scale basins (Steel and Worsley, 1984). The drainage pattern
changed from the west in the Early Triassic to the east from the Middle Triassic.  During this time Svalbard sat
on the peripheries if the largest recorded delta system in Earth's history (Anell et al., 2014; Klausen et al., 2019;



Mørk, 2013). The latest Triassic to middle Jurassic saw much less sedimentation with numerous hiatuses and
changes in drainage (Olaussen et al., 2018; Rismyhr et al., 2019). The late Jurassic to early Cretaceous saw
greater deposition, including regionally important source rock intervals, and change in drainage due to the
opening of the Amerasian Basin (Dypvik and Zakharov, 2012; Koevoets et al., 2018). During the Early
Cretaceous the development of the High Arctic Large Igneous Province is evident from predominantly mafic
dykes and sills in Svalbard (Senger et al., 2014). This likely resulted in major erosion during the late Cretaceous
and early Palaeocene (Jochmann et al., 2019).
In Svalbard and the rest of the Barents Shelf, the Cenozoic geological history is the most important to understand
subsurface fluid flow. In Svalbard, the Eocene was the time of maximum burial (Dörr et al., 2018), while in
much of the Barents Sea  maximum burial and hydrocarbon generation was probably during the late Oligocene to
early Miocene (Henriksen et al., 2011; Faleide et al., 1996). From the Eocene to present major regional uplift of
1 to 3 km has occurred and is ongoing with Svalbard experiencing the greatest uplift magnitudes, hence being
subaerially exposed (Dimakis et al., 1998; Lasabuda et al., 2018). For this study, the most pertinent tectonic
events are of widespread uplift and erosion due to repeated glacial cycles of the past few million years (Dimakis
et al., 1998; Landvik et al., 1998). These recent events are still ongoing, and are the most important with respect
to the migration and leakage of hydrocarbons from deeper traps to the shallow subsurface (Ohm et al., 2008;
Abay et al., 2017).
The prevalence of hydrocarbon shows and gas influxes throughout the stratigraphy can be attributed to the
presence of multiple mature source rocks (Ohm et al., 2019). The marine shales of the Jurassic Agardhfjellet
Formation of the Adventdalen Group and the Triassic Botneheia Formation of the Sassendalen Group are both
regionally extensive and prolific source rocks, responsible for charging the majority of the oil and gas
discoveries in the Norwegian Barents Sea. In addition, organic-rich shales in the Gipsdalen Group (Braathen et
al., 2012) probably represent laterally restricted source rocks. Carboniferous, Cretaceous and Paleogene coal
seams have been exploited in Svalbard's past, with the latter still being produced in the Central Tertiary Basin in
Adventdalen and Barentsburg. These coal seams are widespread and are typically gas-prone and oil-prone source
rocks (Marshall et al., 2015; Uguna et al., 2017).
Numerous sandstones and karstified carbonates provide potential reservoirs throughout Svalbard's stratigraphy,
many of which are direct analogues to proven hydrocarbon reservoirs in the Barents Sea (Nøttvedt et al., 1993).
For this study the most notable reservoirs are the shallow marine sandstone-dominated Lower Cretaceous
Helvetiafjellet and Carolinefjellet Formations (Steel et al., 1981; Grundvåg et al., 2019), and the Triassic-Jurassic
Kapp Toscana Group deltaic to shoreline deposited siltstone and sandstones (Mørk, 1999).
The above-mentioned source rocks are also the best candidates for sealing intervals. In addition, the mudstones
of the late Palaeocene Basilika Formation and Palaeocene-Eocene Frysjaodden Formation also possess potential
sealing properties (Steel et al., 1981). The numerous source rocks, technical oil and gas discoveries, bitumen
stained strata and surface seeps suggest that Svalbard possesses working petroleum systems, though none of the
eighteen exploration wells drilled onshore Svalbard from 1961 to 1994 resulted in commercially viable
discoveries (Senger et al., 2019). Although hydrocarbon accumulations likely first formed tens of millions of




years ago when source rocks were at maximum burial (Magoon and Dow, 2000), subsequent tectonic events
have undoubtedly caused tertiary fluid migration (Abay et al., 2017; Ohm et al., 2008).
The most recent deglaciation of the Barents Ice Sheet from 15 to 10 kya probably caused tilting and hydrocarbon
spillage from existing traps, furthermore glacial and overburden unloading resulted in remigration. Gas,
particularly methane, is the dominant hydrocarbon found in Svalbard due to the prevalence of over-mature or
gas-prone source rocks (Michelsen and Khorasani, 1991; Ohm et al., 2019; Senger et al., 2019) and active
methanogenesis (Hodson et al., 2019). Deglaciation and uplift has reduced confining pressure on subsurface
fluids and led to gas exsolution and expansion. Therefore, the subsurface fluid systems in Svalbard are in a state
of disequilibrium and widespread hydrocarbon migration is likely ongoing at present (Abay et al., 2017).
Evidence of this is manifested as out-of-equilibrium pore pressures (Birchall et al., 2020) and the previously
mentioned surface seeps.

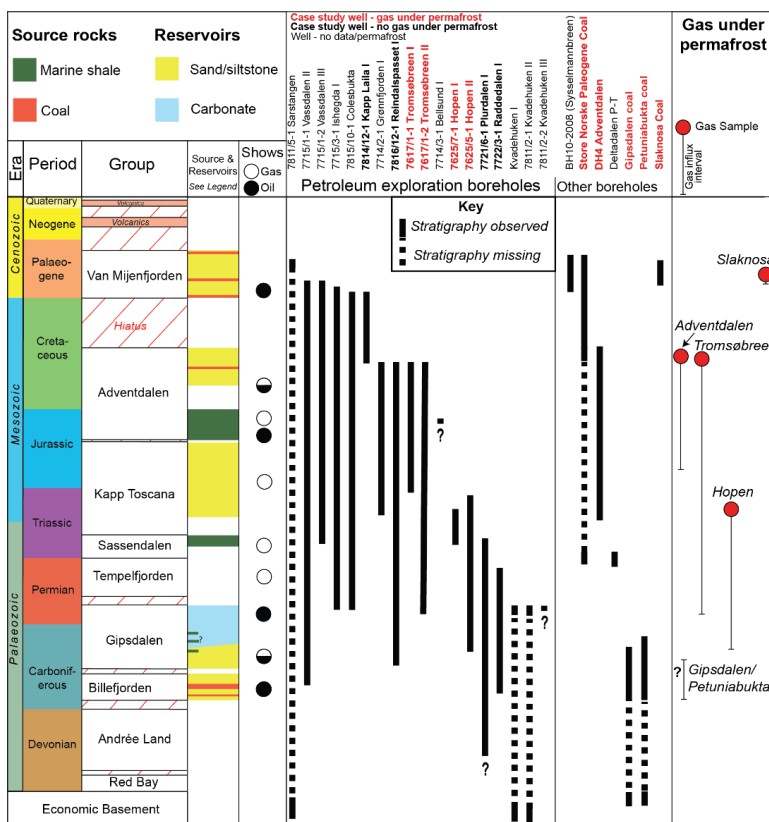

**Figure 2 – The hydrocarbon exploration wells of Svalbard and key coal and scientific boreholes showing the**
**stratigraphy they penetrated, modified from Senger et al. (2019). The base-permafrost, gas-bearing stratigraphy is**
**shown in the right hand column.**
In Svalbard, the timing of hydrocarbon migration and permafrost formation overlap, meaning there is potential
for accumulations to develop beneath the impermeable permafrost. Although numerous hydrocarbon and coal



exploration wellbores penetrate the entire permafrost interval, it has rarely been of interest to the operators
(Senger et al., 2019). However, on detailed inspection of well data, reports, and anecdotal evidence, it is clear
that sub-permafrost gas accumulations have been frequently encountered throughout the archipelago. In
Adventdalen, sub-permafrost free-gas was first documented in 1967 during coal exploration and encountered
again in 1979 (Snsk, 1981). This accumulation was further confirmed, and sampled (Ohm et al., 2019; Huq et al.,
2017), during scientific drilling of the Longyearbyen $CO_2$ Lab between 2008 and 2012. Figure 2 shows the key
wellbores of this study, the stratigraphy they penetrate in Svalbard and whether they encountered gas at the base
of permafrost, which is the main point of discussion in this contribution.

## 183   3 Data and methods

Several decades of coal and petroleum exploration, as well as research drilling, in Svalbard has led to much
anecdotal evidence of gas accumulations beneath permafrost. We have attempted to verify this by analysing data
from boreholes that have penetrated through the permafrost in Svalbard. These boreholes include eighteen
hydrocarbon exploration wells, ten scientific boreholes, eight of which are from the Longyearbyen $CO_2$ Lab
(from two drill sites). Also integral to this study are the somewhat sporadic data, including drilling and
geological reports, from more than five hundred coal exploration boreholes drilled by the local Store Norske
Spitsbergen Kulkompani (SNSK) over a period of nearly a century. We identify where gas accumulations occur
and where these coincide with the base of permafrost, or the first permeable interval below it. One of the major
challenges with these boreholes is that they typically target much deeper stratigraphy and often acquire very
limited petrophysical data in the shallow parts. Typically, only the gamma ray logging tool, which measures the
rocks natural radioactivity, is run in the shallow intervals. The available well data used in this study are presented
in Table 1. Ascertaining the presence of sub-permafrost gas presents several challenges.
Identifying the presence of permafrost is simple and can often be clear from geomorphological features such as
pingos. However, identifying the thickness and base of permafrost is much more challenging (Osterkamp and
Payne, 1981). Table 2 shows the ideal responses of petrophysical and drilling data at the lower permafrost
boundary and the challenges to each method. By far the biggest challenge to petrophysical and drilling data
analysis in Svalbard is due to the low porosity, heterolithic, very rigid and overcompacted rocks (Henriksen et
al., 2011). The nature of the base of permafrost itself is also not well understood, but it is a reasonable
assumption that it a diffuse boundary which adds to the complexity of identifying a permafrost boundary in
petrophysical data alone. Further complications arise from the drilling fluid used and circulated in the wellbores
which was often heated and hypersaline. Nevertheless, it is generally possible to identify the approximate base of
permafrost on a case-by-case basis using all available data. Petrophysical data is robust in identifying lithology
and drilling data is useful in identifying changes in fluid behaviour. When liquid water is encountered it is
obvious evidence of being below the ice-bearing permafrost (though may be below 0°C, depending on salinity).
Other indicators that can help identify the position of permafrost include ice-plug formation within the wellbore,
sudden changes in the character or amount of drill cutting returns and increases in background gas
measurements. The strongest indication of base permafrost occurs where fluid influxes into or out of the
wellbore suddenly occur in thick, normally permeable sandstones. In this situation it is very likely it is due to the
transition of impermeable permafrost to permeable water or gas-bearing rock. Abnormally high pressures at the





apparent base of permafrost are often mentioned in well reports and provide good evidence that the permafrost is
acting as an effective seal.

| | Petrophysics - Start of Data (m MD) | | | | | | Gas Data | | |
| Well | Gamma Ray | Resistivity | Acoustic | Density | Temperature | Cuttings | Gas Shows (Chromatograph) | Fluid Samples | Well Report |
|---|---|---|---|---|---|---|---|---|---|
| Hydrocarbon Exploration | | | | | | | | | |
| 7617/7-1 Tromsøbreen-1 | (Drilling parameters only) | | | | BHT | Surface | Surface | 768 m | Y |
| 7617/7-2 Tromsøbreen-2 | 17 | 350 | 350 | 330 | ? | Surface | Surface | - | Y |
| 7625/7-1 Hopen-1 | 3.5 | - (SP logged) | - | - | BHT | Surface | Surface | c. 150 m | Y |
| 7625/6-2 Hopen-2 | Surface | 349 | 349 | 638 | Surface | Surface | Surface | - | Y |
| 7714/2-1 Grønnfjorden | not logged | | | | | Cored | - | - | Y |
| 7714/3-1 Bellsund | ? | | | | | | | | N |
| 7715/1-1 Vassdalen-2 | Surface | 17 | - | - | - | - | - | - | N |
| 7715/1-2 Vassdalen-3 | - | - | - | - | - | - | - | - | N |
| 7715/3-1 Ishøgda | Surface | Surface | Surface | Surface | Surface | Surface | - | - | N |
| 7721/6-1 Plurdalen | 5 | 83 | 5 | 83 | Surface | Surface | Surface | Water at 500 m | Y |
| 7722/3-1 Raddedalen | Surface | 5 | 591 | 593 | 5 | Surface | Surface | - | Y |
| 7811/2-1 Kvadehuken-1 | not logged | | | | | Cored | - | - | N |
| 7811/2-2 Kvadehuken-2 | not logged | | | | | Cored | - | - | N |
| - Kvadehuken-0 | Shallow, no data | | | | | | | | |
| 7811/5-1 Sarstangen | 30 | 615 | | | BHT | Surface | 260m | - | Y |
| 7814/12-1 Kapp Laila | Surface | - (SP logged) | | | | 24 m (partial recovery) | | | Y |
| 7815/10-1 Colesbukta | Surface | 41 | 1467 | - | - | - | - | - | N |
| 7816/12-1 Reindalspasset | From surface | 22 (induction) | 22 | 22 | 17.4 | Suface | 20 m | - | Y |
| Selected Coal Boreholes | | | | | | | | | |
| 1967-1 Adventdalen | Cored (not logged) | | | | | | Y | - | Y |





| Borehole | | | | | | | | | |
|---|---|---|---|---|---|---|---|---|---|
| 1979-10 Adventdalen | | | | | | | - | - | Y |
| 1979-11 Adventdalen | | | | | | | - | - | Y |
| DDH1B Gippsdalen | | | | | | | - | - | Y |
| 1982-20 | No data | | | | | | - | - | 1982 drilling summary |
| Gruve 7 - H1 | | | | | | | Y | - | 1979 drilling summary |
| 1981-02 | | | | | | | - | - | 1981 drilling summary |
| 1981 (Platåberget) | | | | | | | - | - | 1981 drilling summary |
| 1981-05 Breinosa | | | | | | | - | - | 1981 drilling summary |
| 1981-06 Breinosa | Cored (not logged) | | | | | | - | - | 1981 drilling summary |
| 1979-1 Reindalen | | | | | | | - | - | Y |
| 1990-12 Slaknosa | | | | | | | - | - | Y |
| Scientific Boreholes | | | | | | | | | |
| DH1 | 3 | 3 | 9 | - | Surface | Cored | - | | Y |
| DH2 | 10 | 10 | 10 | - | Surface | Cored | - | - | Y |
| DH3 | Cored: not logged | | | | | Cored | - | - | Y |
| DH4 | Surface | 440 | 440 | - | Surface | Cored | - | Through out | Y |
| DH5r | 3 | - | 100 | - | Surface | Cored | - | Below 645 m | Y |
| DH6 | Cored: not logged | | | | | Cored | - | | Y |
| DH7a | Cored: not logged | | | | | Cored | - | Below 645 m | Y |
| DH8 (Shallow) | Cored: not logged | | | | | Cored | - | | Y |
| BH10-2008 | Surface | 67 | 48 | Surface | - | - | - | - | Y |
| Janssonhaugen (temperature) | - | - | - | | Surface | - | - | - | N |

**Table 1 – Data availability and intervals recorded for the permafrost penetrating boreholes.**



Direct temperature data from thermometers used in conjunction with wireline logging tools is common from
hydrocarbon exploration wells. However these were rarely allowed to reach thermal equilibrium with the
surrounding formations following drilling and fluid circulation. Therefore accurate absolute temperature
measurements are rare, though temperature trends (e.g. inflection points) can be used more qualitatively to
estimate base permafrost. Wells monitored over longer time periods, such as the scientific boreholes in
Adventdalen (Isaksen et al., 2000; Olaussen et al., 2019; Juliussen et al., 2010) are relatively rare, but provide
much more reliable and precise temperature data.
Identifying the presence of gas is relatively simple and, although petrophysical data is generally not helpful in
shallow sections for fluid discrimination. Reliable evidence comes from influxes of gas into the wellbore which
has been sampled from wells in Adventdalen, Tromsøbreen and Hopen (Senger et al., 2019). Elevated
background gas is another good indicator of sub-permafrost gas and is measured in drilling fluids returning to the
surface and extracted by a "gas trap". This method typically identifies in-place dry gas accumulations or gas that
has exsolved from fluid on its way to the surface due to pressure decline. However, these measurements do not
detect gas that remains dissolved in formation water. Gas from drilling mud is also impacted by a variety of
factors (Marum et al., 2019), including drilling rate, drilling mud type and, perhaps the most pertinent,
temperature; low temperatures can cause heavier hydrocarbons to condense and avoid detection, it also causes
drilling fluids to become more viscous, further inhibiting gas extraction.

| Log Type | Property Measured (Units) | Idealised Permafrost Response | Complicated by |
|---|---|---|---|
| **Petrophysical Data** | | | |
| **Gamma Ray** | Radioactivity of rocks (API) | No Response but useful in determining lithology. | N/A |
| **Acoustic (Sonic)** | Seismic velocity of rocks and fluids within. Measured in slowness (microsecond per foot) | Faster velocities (lower slowness) in icebound intervals. | Overcompacted, dense and rigid rocks. Low porosity and heterolithic rocks. |
| **Resistivity** | Resistivity of rocks and fluids within. | High resistivity in permafrost becoming low in water bearing interval. | Resistive hydrocarbons below permafrost. Fresh water below. Low porosity rocks. Clay rich and heterolithic rocks. |
| **Density** | Density of rocks and fluids within. | Decreased density in ice-bearing intervals | Low porosity. Heterolithic rocks (fluid response is generally overwhelmed by lithological response). |
| **Temperature** | Temperature of fluid in borehole at a given depth. | 0°C or lower in permafrost interval. | Measures wellbore fluid, not fluid within formation. Drilling fluid circulates and is often heated. Requires a long time to equilibrate to formation. |



| Fluid Sampling | Pressure and fluid properties. | Qualitative - shows fluid phase and type. Abnormal pressures indicate a vertical barrier or seal. | Low permeability (including permafrost ice). Limited to few points in well. Shallow samples rarely of interest. |
|---|---|---|---|
| **Drilling Parameters** | | | |
| **Fluid Influx** | Fluid entering wellbore (often flowing to surface) | Indicates transition from impermeable to permeable zone. | Exact depth of influx is uncertain. |
| **Background Gas** | Measures levels and composition of gas returned with drilling fluid at the surface. Does not measure dissolved gas. (Percentage or parts per x) | Indicates transition from impermeable to permeable zone. | Varies depending on drilling rate, permeability, drilling mud type. Exact depth/formation of gas origin is uncertain. |
| **Rate of Penetration (ROP)** | The rate the drill bit penetrates the ground (ft. or m per hour) | A rapid rise in rate of penetration when transitioning from ice-bound to unbound rock. Best used with Weight-on-bit (1000 lbs) measurement. | Well-cemented, compacted and hard rocks. Rate can depend on external factors including drill bit condition. |
| **D-exponent** | Extrapolation of numerous drilling parameters to estimate pore-pressure. | May identify anomalously high drilling rate (or pressure) at base permafrost. | Well cemented, compacted and hard rocks. Heterolithic rocks. Largely qualitative. |

**Table 2 – The petrophysical and drilling parameters that may identify permafrost and its base. Idealised responses**
**are shown and typically identify the transition from ice to water. The final column shows the complicating factors, all**
**of which are applicable to Svalbard. Perhaps the most pertinent complication to Svalbard is that the geology is**
**comprised of well cemented, compacted, hard rocks.**
In addition to well data, we also include data from published scientific studies including isotope (Huq et al.,
2017), thermobaric (Isaksen et al., 2000; Betlem et al., 2018; Betlem et al., 2019), geophysical (Beka et al.,
2017; Johansen et al., 2003) and geochemical (Leythaeuser et al., 1984; Ohm et al., 2008) analyses. We also
analysed Russian published literature for areas operated by Trust Arktikugol (Lyutkevich, 1937; Verba, 2013).
For Adventdalen, Tromsøbreen and Hopen we integrated all relevant data (summarised in Betlem et al., in
review; and references therein) in order to calculate the gas hydrate stability zone and permafrost extent for the
targeted study areas according to the workflow outlined in Betlem et al. (2019) and further refined in Betlem et
al. (in review). The workflow assumes steady-state conditions and implements structure-I gas hydrate phase
boundary curves generated through the HWHYD modelling software (Masoudi and Tohidi, 2005).



# 4 Results

## 4.1 Evidence of Permafrost

Figure 3 shows which boreholes encounter permafrost, their elevation, and the depth to the base of permafrost.
Although it does not preclude its existence, there is no clear evidence of permafrost it in the hydrocarbon
exploration wells on the west coast. On the shoreline of Isfjorden, the Kapp Laila and Adventdalen (DH1 and
DH2) wells show evidence of a thin permafrost interval, although it may not be ice-bearing. Wells on the east
coast of Spitsbergen at Tromsøbreen and on the southern beach of Hopen provide strong evidence of a thicker
permafrost top seal even in coastal locations. Wells further inland, including the majority of coal exploration
boreholes, unsurprisingly show evidence of thicker permafrost.

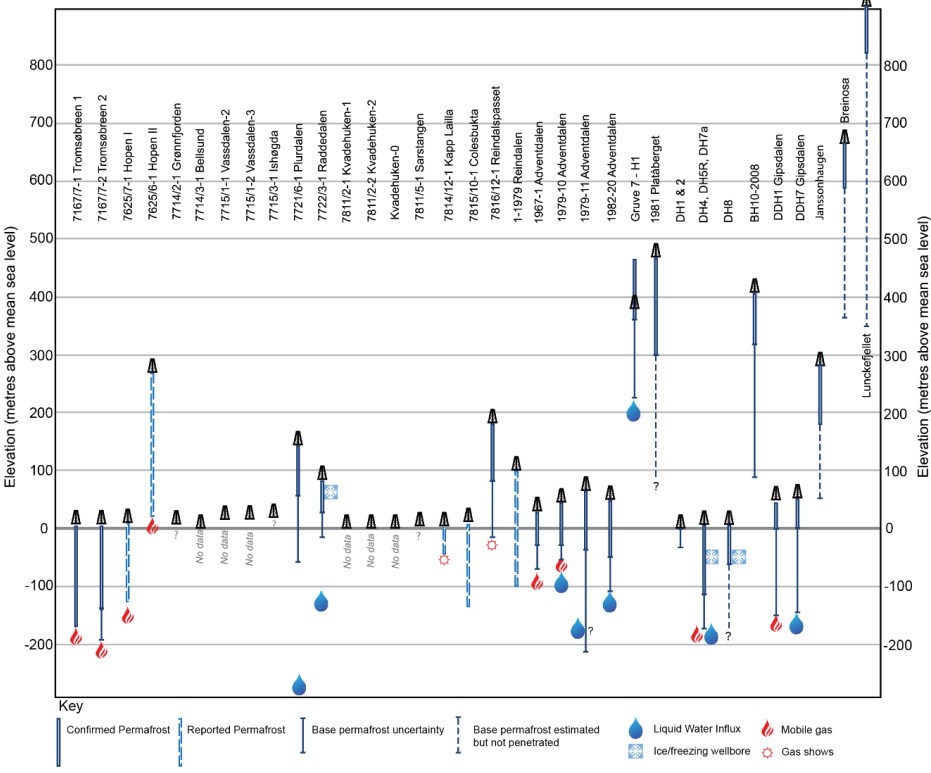

**Figure 3 - A plot of wells in this study showing their elevation and the depth to base permafrost. Solid well path
outlines show where data analysed in this study confirms the presence of permafrost while dashed outlines represent
where base permafrost has been reported but data is not available. For the Breinosa wellbore, which shows -7.8° at its
coldest point at 78 m (and a TD at 90 m) (Juliussen et al., 2010), we extrapolated the base permafrost the local
geothermal gradient of 35°/km (Betlem et al., 2018; Isaksen et al., 2000). Borehole locations are shown in Fig. 1.**

Table 3 shows occurrences of where gas has and has not been encountered at the permafrost base. The wells in
Adventdalen, Tromsøbreen, Hopen and Gipsdalen all indicate gas accumulation at the base of permafrost and all





but the latter are discussed in detail here. Reindalen, Kapp Laila and the Plurdalen and Raddedalen wells on
Edgeøya are also of particular interest and discussed further because they show good evidence of permafrost, but
do not appear to encounter gas accumulations below it.

| Well | Evidence for Gas Under Permafrost | Tentative/Shows | Permafrost but no gas |
|---|---|---|---|
| *Hydrocarbon Exploration* | | | |
| 7617/7-1 Tromsøbreen-1 | • | | |
| 7617/7-2 Tromsøbreen-2 | • | | |
| 7625/7-1 Hopen I | • | | |
| 7625/6-1 Hopen II | • | | |
| 7714/2-1 Grønnfjorden | | | |
| 7714/3-1 Bellsund | | | |
| 7715/1-1 Vassdalen-2 | | | |
| 7715/1-2 Vassdalen-3 | | | |
| 7715/3-1 Ishøgda | | | |
| 7721/6-1 Plurdalen | | | • |
| 7722/3-1 Raddedalen | | | • |
| 7811/2-1 Kvadehuken-1 | | | |
| 7811/2-2 Kvadehuken-2 | | | |
| Kvadehuken-0 | | | |
| 7811/5-1 Sarstangen | | | |
| 7814/12-1 Kapp Laila | | • | |
| 7815/10-1 Colesbukta | | | |
| 7816/12-1 Reindalspasset | | | • |
| *Coal* | | | |
| 1967-1 Adventdalen | • | | |
| 1979-10 Adventdalen | • | | |
| 1979-11 Adventdalen | | | • |
| 1982-20 Adventdalen | | | • |
| Gruve 7 - H1 Adventdalen | | | • |
| DDH1B Gippsdalen | • | | |
| 1979-1 Reindalen | | • | |
| 1981 Platåberget | | | |
| 1981-Breinosa | | | |
| Lunckefjellet | *TD above base permafrost* | | |
| Ispallen | | | |
| 1990-12 Slaknosa | • | | |
| *Scientific Wellbores* | | | |
| DH1 | | | |
| DH2 | | | |
| DH3 | | | |
| DH4 | • | | |
| DH5r | • | | |



| | | |
|---|---|---|
| **DH6** | | |
| **DH7a** | | |
| **DH8 (Shallow)** | | |
| **BH10-2008** | | • |
| *Janssonhaugen* | *TD above base permafrost* | |

**Table 3 – Wells showing where gas is and is not present at the base of permafrost. Wells without a bullet either**
**contain no permafrost or no relevant data.**

## 4.2 Case Studies: confirmed sub-permafrost gas

### 4.2.1 Adventdalen

Svalbard's largest settlement, Longyearbyen, is located in Adventdalen (Fig. 4), and one of the better studied
areas of Svalbard (Hodson et al, 2020; Hornum et al, 2020; Johansen et al., 2003; Beka et al., 2017; Olaussen et
al., 2019 and references therein). The wells of the Longyearbyen $CO_2$ Lab and coal exploration boreholes of
SNSK both show the presence of gas beneath the permafrost in Adventdalen (Fig. 5) provides a correlation panel
of these wellbores.

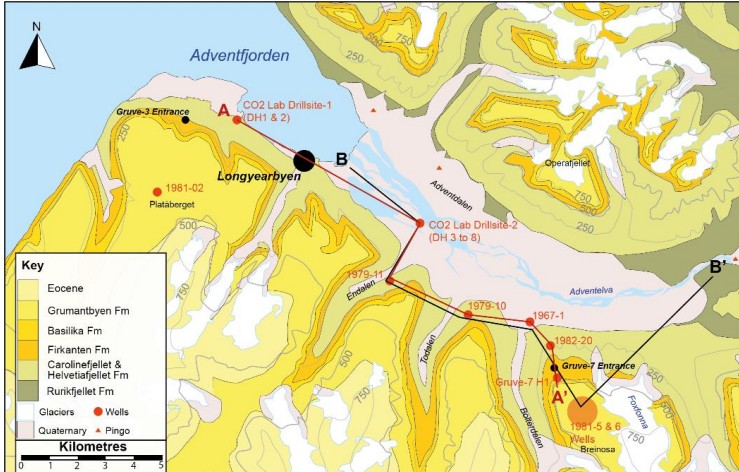

**Figure 4 - A Geological Map of Adventdalen showing some of the youngest stratigraphy exposed in Svalbard (base**
**map data courtesy of © Norwegian Polar Institute). The profile A to A' represents the well correlation in Fig. 5 and B**
**to B' the modelled permafrost profile in Fig. 6.**
At the near-coast drillsite-1 of the Longyearbyen $CO_2$ Lab wells temperature data from DH1 and DH2 indicate a
thin permafrost interval with the base at approximately 20-30 m (Beka et al., 2017). Although sub-zero
temperatures were recorded at this site, the presence of ice is strongly dependent on the pore-fluid salinity. At
drillsite-2, wellbores DH3 and DH4 encountered overpressured water at the base permafrost. DH4 and DH5R
also encountered significant natural gas with this water kick and it was collected in gas bags for sampling (Ohm





et al., 2019; Huq et al., 2017). Temperature logs from well DH4 suggest base permafrost from 150 to 200 m
depth but given the drilling fluid losses and mud circulation there is considerable uncertainty in this data. Cores
from nearby wells DH6 and DH7A also show elevated methane levels at this depth. The water and gas influxes
occur somewhere towards the middle (i.e. not top of the reservoir) of the sandstone dominated Helvetiafjellet
Formation. Figure 6 is the modelled permafrost thickness in Adventdalen which shows good agreement with the
independent well data observations.







**Figure 5 – A Well correlation of base permafrost with all available geological, drilling and petrophysical data. The location of the correlation is shown in Fig. 4. The wells in this section highlight the somewhat sporadic nature of data availability over the shallow, permafrost bearing intervals.**


In the same area, hundreds of coal boreholes, drilled by SNSK over the decades, have penetrated the permafrost interval, although data for these is more fragmented. Well 1979-11 was drilled approximately two kilometres south of Longyearbyen CO$_2$ Lab drillsite-2 in Endalen. This well encountered water influxes with no mention of gas, although no depths are stated in the report (Snsk, 1980, 1981). Well 1979-10, two kilometres to the



southeast in the neighbouring valley Todalen encountered methane-rich gas overlying inflowing water at the
base of permafrost at a depth between 150 to 200 m (Snsk, 1981, 1982b; Leythaeuser et al., 1984). Well 1967-1,
approximately three kilometres east and geologically updip of 1979-10, reached a depth of 106 m where a gas
accumulation was encountered (Snsk, 1981).  This well was also the subject of considerable interest by SNSK
who investigated the potential of producing the gas commercially. Well 1982-20, approximately one kilometre
southeast of 1967-1, at the base of Breinosa and the coal mine Gruve-7, did not encounter gas and took water
influxes of 33-42 litres per minute at approximately 150 m at the base of permafrost (Snsk, 1982a). Another
reported well, named only "first water well", (Snsk, 1982a) in the same area flowed from the same interval at 40-
50 litres per minute. Water from these two wells had a measured chloride concentration of 1500 ppm (Snsk,
1982a).  A well drilled inside Gruve-7 at approximately 380 m AMSL encountered liquid water at 154 m depth.

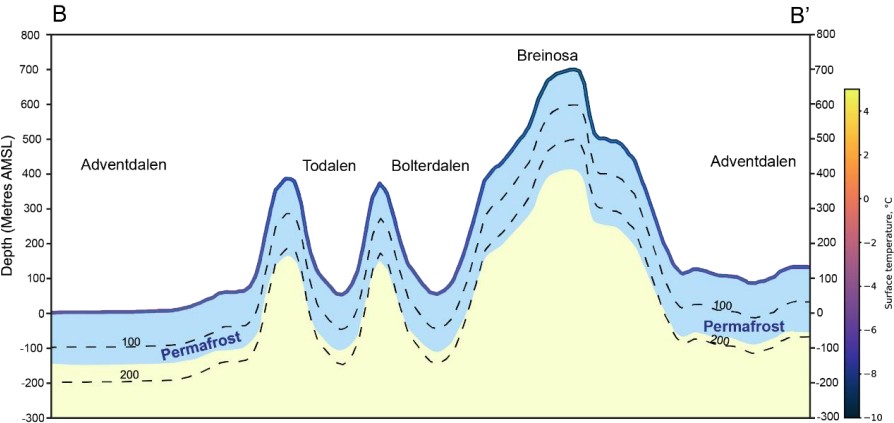

**Figure 6 - Modelled permafrost thickness through Adventdalen with the profile shown in Fig. 4. The model**
**parameters are discussed in the methods section but note that the permafrost interval is entirely based on temperature**
**rather than ice thickness or presence.**
Well 1967-1 and 1979-10 most likely encountered the same gas accumulation, while well 1982-20 encountered
permafrost over the same stratigraphic interval and well 1979-11 is probably down-dip of the gas-water
interface. Intermittent flow from the 1967-1 well was monitored between October 1967 and July 1975 (Snsk,
1981). The first year of this production was continuous and monitored as shown in Fig. 7A. An initial wellhead
gas pressure of 14 bar was recorded (Snsk, 1981) with relatively slow pressure and production decline over time.
This indicates the gas accumulation is in the order of millions of cubic-metres and has well connected pressure
support. If the aquifer pressure is known then the length of the methane gas column can be calculated from this
pressure. It is clear the aquifer is not at hydrostatic pressure from the surface due to the repeated influxes and
water flow from wells 1979-10, 1979-11 and in the $CO_2$ lab research boreholes. Unfortunately these pressures
were not measured.
SNSK commissioned flow analysis work to be carried out on the 1967-1 well in July 1975 and the results of
these two test runs are shown in Fig. 7B. Here it is clear the well responded to pressure drawdown. However,
flow rates were still significantly lower than those recorded over the first year. Figure 7C shows the pressure



build-up when the well was shut in (effectively closed from the atmosphere) between the two test runs. The
quick return to pre-drawdown pressures indicates, somewhat unsurprisingly, a good natural pressure support in
the well. Ultimately the gas here was deemed by Statoil, and consequently SNSK, to be an uneconomic
accumulation locally trapped by permafrost (Snsk, 1981).

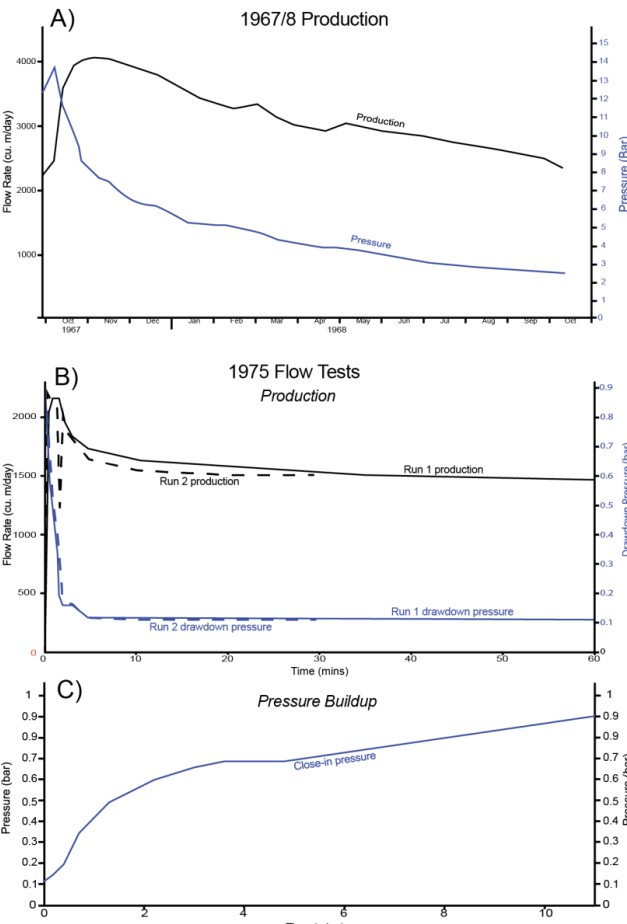


**Figure 7 – Production tests on the 1967-1 well in Adventdalen where was produced and flared. A) Gas production and**
**pressure depletion for the first year of production. B) An oilfield-standard production test of production and pressure**
**drawdown carried out by Statoil in 1975. The relatively fast flattening of the curves suggests stable flow and strong**
**pressure communication in the reservoir. C) Pressure build up following shut-in of the well also indicating**
**appreciable fluid communication.**
A drilling summary (Snsk, 1982a) documents two wells drilled at approximately 400 m above mean sea level
(AMSL) on Platåberget, on the southern side of Adventdalen. They both report total drilling fluid losses at 160 -
170 m MD with no record of gas influxes. This is within the permafrost interval based on the presence of
permafrost in the coal mine, Gruve-3, some 200 m below the surface. This demonstrates that the permafrost
interval here is permeable. Similarly, on Breinosa, where the Gruve-7 mine is situated some fifteen kilometres to





the east, wells 81-05 and 81-06 both encountered total fluid losses at a similar depth of 170m (Snsk, 1982a), well
within the permafrost interval (Juliussen et al., 2010). The mine itself is situated entirely within the permafrost
interval  and has suffered from meltwater influxes from the overlying cold-based glacier, Foxfonna, on numerous
occasions (Christiansen et al., 2005). Similar losses occurred in several intervals between 106 and 196 m in well
19-2011 on Operafjellet, a plateau on the northern side of Adventdalen (Snsk, 2011a). Freezing in the wellbore at
132 m indicates at least some, if not all, of these losses occurred in the permafrost interval.
Five pingos are situated along the northern edge of Adventdalen. Four of them provide active migration
pathways through the permafrost leading to the discharge of brackish springs and high concentrations of methane
(up to and marginally exceeding the solubility limit of 41 mg L-1) (Hodson et al., 2020). At the easternmost
pingos, the chloride concentrations and the d$^{13}$C isotopic composition of both the methane and dissolved $CO_2$ are
remarkably similar to those described in the wellbore records above.
**4.2.2 Tromsøbreen**
Two hydrocarbon exploration wells, Tromsøbreen-I (7617/1-1) and Tromsøbreen -2 (7617/1-2), were drilled at
Haketangen in south-eastern Spitsbergen in 1977 and 1988, respectively. Both were drilled in nearly the same
coastal location at 6 m AMSL, near the terminus of the Tromsøbreen glacier (Senger et al., 2019).
The wells primarily targeted the Jurassic-Triassic sandstones in an anticline trap mapped on the surface to the
west (Fig. 8A) with the wells planned to be slightly deviated to intersect this at the prospect depth (Norsk Polar
Navigasjon a/S, 1977b, a; Polargas Prospektering Kb, 1988). The outcrops in this area are predominantly the
Carolinefjellet and Helvetiafjellet sandstones, though older stratigraphy outcrops to the west near the WSFTB
hinterland. Unfortunately, gamma ray was the only petrophysical data acquired over the shallow intervals,
though gas chromatography, drilling parameters and drilling and well reports provide a good indication of the
subsurface.
Both wells suffered major drilling problems at the apparent base of permafrost at 179 m. The permafrost interval
showed no permeability and in Tromsøbreen-1 took 45 days (the entire wellbore took 90 days) to successfully
drill through (Norsk Polar Navigasjon a/S, 1977b). Both wells suffered major drilling fluid losses into the
formation; this was measured in Tromsøbreen-1 at 150 to 200 barrels (24 to 32 cubic metres) of drilling mud
(Norsk Polar Navigasjon a/S, 1977b). At the same time as drilling fluid was lost from the wellbores, gas influxes
into the both wells also occurred. Indeed, measurements show significant natural gas from this point
continuously until the Triassic stratigraphy including a gas kick at 960 m in Tromsøbreen-1. Immediately after
the first gas influx lost circulation material was used to remedy drilling fluid losses. Lost circulation material is
used to plug cavities in the formation to prevent further losses, it also renders the mud gas traps unusable over
the interval it is used, as is shown by "LCM in mud" in Fig. 8B. The shallowest gas sample was taken at 768 m
and comprised predominantly methane and is discussed later in this section. Gas observed throughout the
intervals of both wellbores was deemed by the operator as important enough to plan a third well approximately
one kilometre to the north (Polargas Prospektering Kb, 1988), although it was never drilled.

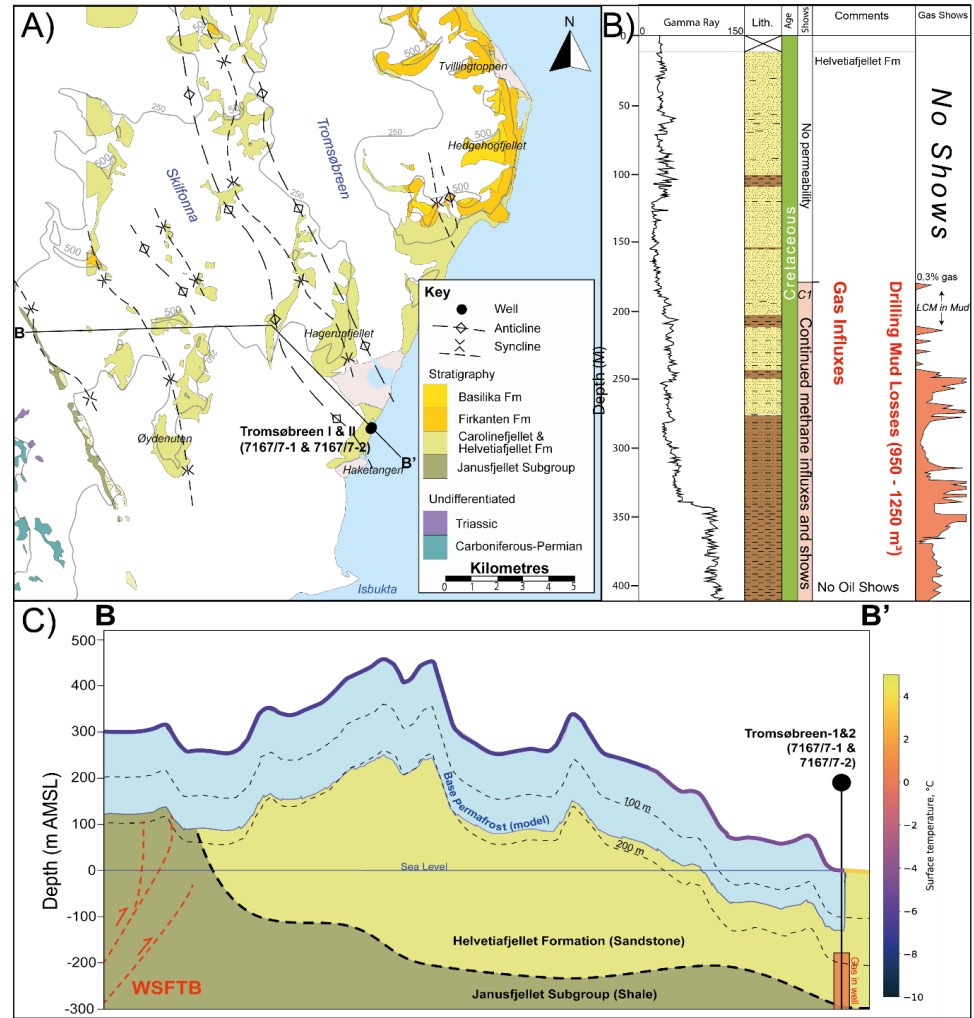

**Figure 8 – A) Geological map from Tromsøbreen redrawn from Birkenmajer et al. (1992). B) All available data over the shallow intervals at Tromsøbreen combined from the two wells. The petrophysical, lithological and gas data is from 7617/7-1 (Tromsøbreen I) while 7617/7-2 (Tromsæbreen II) recorded very little data over the shallow intervals, though did corroborate drilling fluid losses and gas influxes at the same depths. C) Cross-section (shown in A) of modelled permafrost thickness and the important reservoir and sealing formations, and inferred faults of the West Spitsbergen Fold and Thrust Belt (WSFTB) based on outcrop data (Birkenmajer et al, 1992).**

Based on bottom-hole temperatures in both wells, the Tromsøbreen area has an extremely high geothermal gradient, with averages for Tromsøbreen-2 suggesting 43°C/km and Tromsøbreen-1 indicating 52°C/km (Betlem et al., 2018). Fig. 8C shows a simple modelled permafrost thickness using this geothermal regime and surface temperatures. The apparent permafrost encountered in the wellbores has a discrepancy with the steady-state assumption model of approximately forty metres.



### 4.2.3 Hopen

The island of Hopen is 34 km long and 0.5-2.5 km wide and is comprised almost entirely of the heterolithic

Triassic De Geerdalen Formation, which is approximately 650 m thick here (Lord et al., 2014). Two wells were

completed on Hopen, Hopen-1 (7625/7-1) and Hopen-2 (7625/5-1), drilled in 1971 and 1973, respectively

(Senger et al., 2019). Hopen is one of the few cases where the operator took interest in the sub-permafrost gas

accumulation and sampled it. Hopen-1 was drilled on the southern beach while Hopen-2 was drilled in the

highlands in the northern part of the island (Fig. 9).

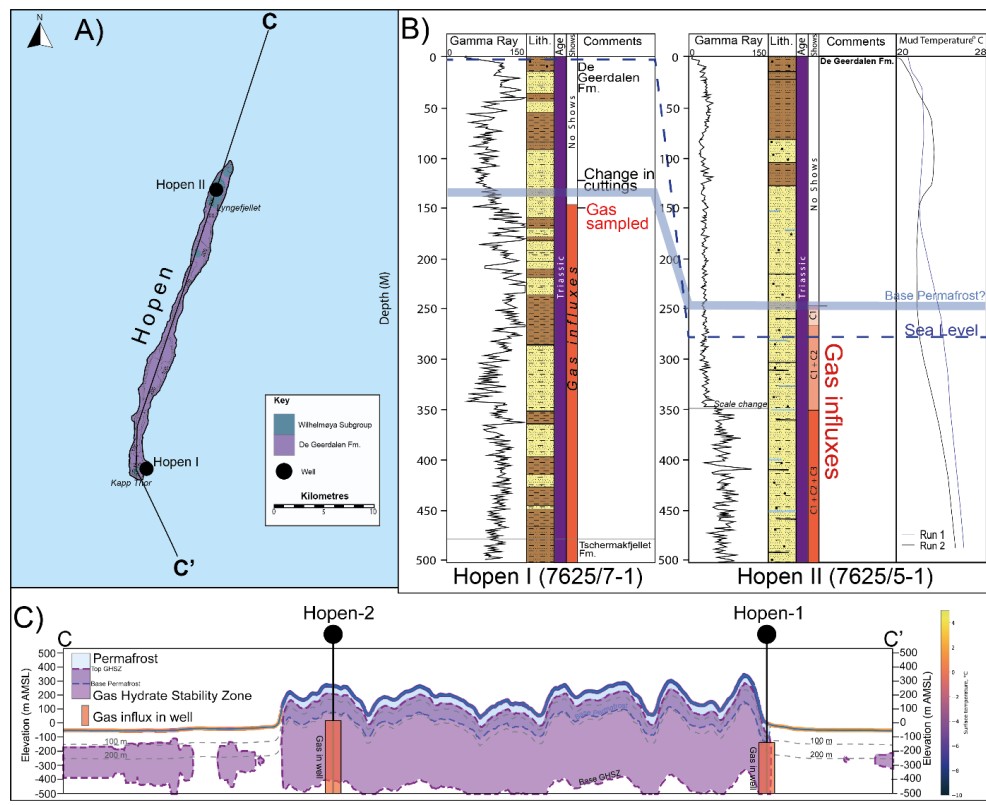

**Figure 9 – A) Geology and outline of the island of Hopen based on Lord et al. (2014) with the map location shown in**
**Fig. 1). Profile C-C' is shown in C) B) Petrophysical and lithological information from the respective wells. Gas**
**samples were taken in Hopen I while a chromatograph in the mud traps was used in Hopen II. The muted gamma ray**
**response in the upper 350 m of Hopen II is probably due to the recording through casing. C) Cross sectional (shown in**
**A) of Hopen showing the modelled permafrost and gas hydrate stability zones. Geology is not shown but the section**
**comprises almost entirely of the heterolithic sand, siltstone and shales of the De Geerdalen Formation.**

Both wells sustained gas influxes attributed to the base of permafrost (Norske Fina a/S, 1971a, b, 1973b, a). In

terms of petrophysical data, operations at Hopen-1 only acquired gamma ray data over the uppermost interval

while Hopen-2 gathered gamma ray and temperature data. However, it is important to note that the wells also

used heated drilling fluids to prevent freezing in the permafrost interval so absolute temperature values in this



section are of limited use. Gas samples were taken in the Hopen-1 well from approximately 150 m while at
Hopen-2 a gas chromatograph was used in the drilling mud traps. Based on temperature data from these wells the
geothermal gradient of Hopen is 25-34°C/km (Betlem et al., 2018).
Hopen-1 was drilled on the southern coast and encountered a gas kick at approximately 150 m which was
deemed significant enough to be sampled. This gas is much heavier in composition than the gas encountered in
Adventdalen and Tromsøbreen and is discussed later in this section. The wellsite geologist noticed an abrupt
change in the cuttings characteristics, but not their lithological composition, at 138 m (Norske Fina a/S, 1971b)
which was attributed to the base of permafrost. Gas was recorded from permafrost base to the bottom of the well
at 908 m (Norske Fina a/S, 1971a, b).
Hopen-2 was drilled approximately 30 km further north on Lyngefjellet. Elevated gas readings in returning
drilling mud were recorded from approximately 250 m (approximately 30 m AMSL) with no apparent changes in
cutting lithology.

### 4.2.4 Slaknosa and Kapp Amsterdam

In 1990 SNSK's 399 m deep coal exploration wellbore 1990-12 encountered a gas blowout at around
approximately 550 m AMSL on Slaknosa plateau on the southern edge of Reindalen (Snsk, 1991). The blowout
exerted enough force to blow rocks, gravel and gas out of the wellbore. Little data remains from this wellbore
and the exact interval of the gas kick is unknown although it was hypothesised to originate from fractured
intervals having migrated from nearby coal seams.
Kapp Amsterdam is a cape close to the mining settlement of Svea. It is comprised of a glacial moraine
approximately 600 years old (Kristensen et al., 2009). In 1986 a methane blowout occurred when drilling
through these deposits at a depth of 33.5 m (Snsk, 1986). According to the drilling report, thermistors were
placed in the wellbore with suggestion that permafrost was acting as a top seal (Snsk, 1986).

### 4.2.5 Gipsdalen

There are very limited data from Gipsdalen, but a single drilling summary report (Senger et al., 2019; Snsk,
1982b, 1979) shows that eight coal exploration wells were drilled in the area. One of these, DDH7, encountered
overpressured water at the base of permafrost at 200 m in either Permian or Carboniferous rocks. The basis for
determining base permafrost is not given but the report states a depth of 300 m was expected prior to drilling.
Another well, DDH1, suffered a gas kick from the same apparent interval. The wellhead pressure from the water
influx in DDH7 was 23 bar while no flow rates were recorded. If the aquifer overpressure is artesian, it equates
to a hydraulic head at approximately 330 m AMSL which may correlate to recharge from heavily glaciated areas
to the east.



## 4.3 Case studies: Permafrost present with no trapped gas

### 4.3.1 Kapp Laila and Colesbukta

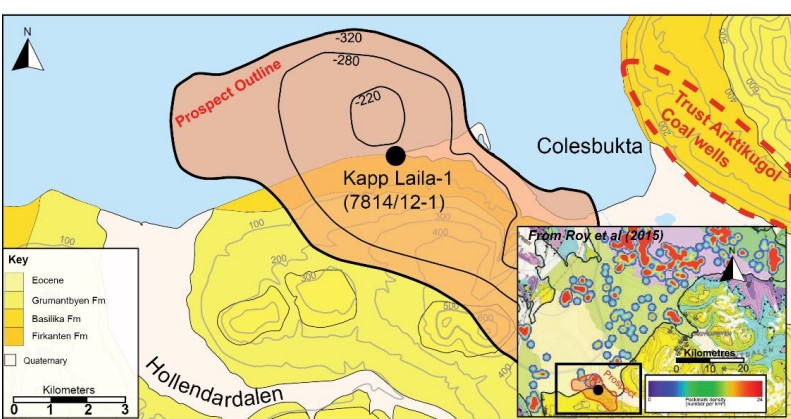

**Figure 10 - Map of the Kapp Laila area (based on SNSK, 1994 and data courtesy of © Norwegian Polar Institute) with the SNSK prospect and well location shown. Inset is a map of pockmarks on the seafloor of Isfjorden from Roy et al (2015). A high concentration of pockmarks on the seabed apparently overlies the crest of the prospect. The map location is shown in Fig. 1.**

Given the coastal location, permafrost is considered to be relatively thin, if present, and is almost certainly absent further offshore (Majewski and Zajaczkowski, 2007). Data from the Trust Arktikugol Colesbukta hydrocarbon exploration well (7815/10-1) is very limited though gas was reported to flow from deeper Triassic intervals (Senger et al., 2019). The SNSK Kapp Laila hydrocarbon exploration well (7814/12-1) does document some fifty metres of permafrost; although it is unclear on what information this is based on (Snsk, 1994). The well and prospect locations are shown in Fig. 10, interesting the crest of the prospect coincides with a cluster of pockmarks offshore (Roy et al., 2015). Gas shows in the form of dull yellow fluorescence were also documented at 44-52 m (Fig. 11), which coincides with the stated permafrost depth. It is important to note that fluorescence shows are not unequivocal proof of hydrocarbons and that yellow fluorescence can also be caused by dolomite and aragonite, although there is no evidence of these minerals in this interval. We have also identified methane seeps through a pingo system approximately 8 km to the east at Trodalen which are the subject of ongoing research in the area.

Trust Arktikugol coal boreholes from the early twentieth century apparently typically encountered permafrost at 100 m depth (Lyutkevich, 1937). Though no specific wells are mentioned, the approximate location of these boreholes is highlighted in Fig. 10. These wells also encountered artesian water at depths of 229-339 m which flowed at 110 litres per minute (Lyutkevich, 1937).



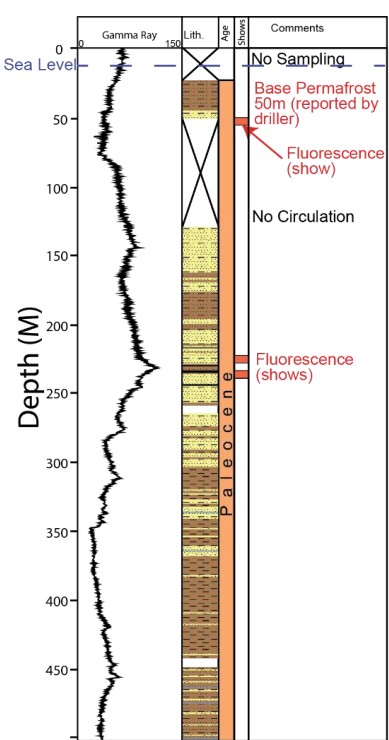


**Figure 11 - The Kapp Laila well with all available data and geological and drilling comments. Minor gas shows occur**
**at 50 m depth which coincides with the stated base of permafrost (SNSK, 1994).**
**4.3.2 Reindalen**
The Reindalspasset well (7816/12-1) was drilled in 1991 by SNSK and Norsk Hydro and was the first well to
target a prospect identified by seismic data (Senger et al., 2019). It also has a good set of petrophysical data over
the permafrost-bearing interval. The Reindalen well is situated on the eastern fringes of the Central Tertiary
Basin (Fig. 12A) but its primary target was a deeper rotated fault block of the Carboniferous Billefjorden fault
zone. Well data suggests a geothermal gradient of 31°C/km (Betlem et al., 2018). Another observation of this
area is the prevalence of pingos in the valley to the east and west which are indicative of migration pathways
through the permafrost, some of which also exhibit methane seepage.
The well data shown in Fig. 12B demonstrates the challenges in identifying permafrost from petrophysical data,
particularly in Svalbard where rocks are typically overcompacted. The rapid resistivity cycling in upper parts is
likely due to thawing of permafrost and intermittent invasion of highly conductive, saline drilling fluids, though
this is purely speculative. There are no major indicators in the acoustic data. Indeed, in both acoustic and
resistivity data, probably due to low porosity in the permafrost bearing zone. The first good quality sandstone
intervals at around 170 m do possess low resistivities which are probably indicative of liquid water. Although the
2D seismic line is of good quality the permafrost does not manifest itself for reasons previously discussed.

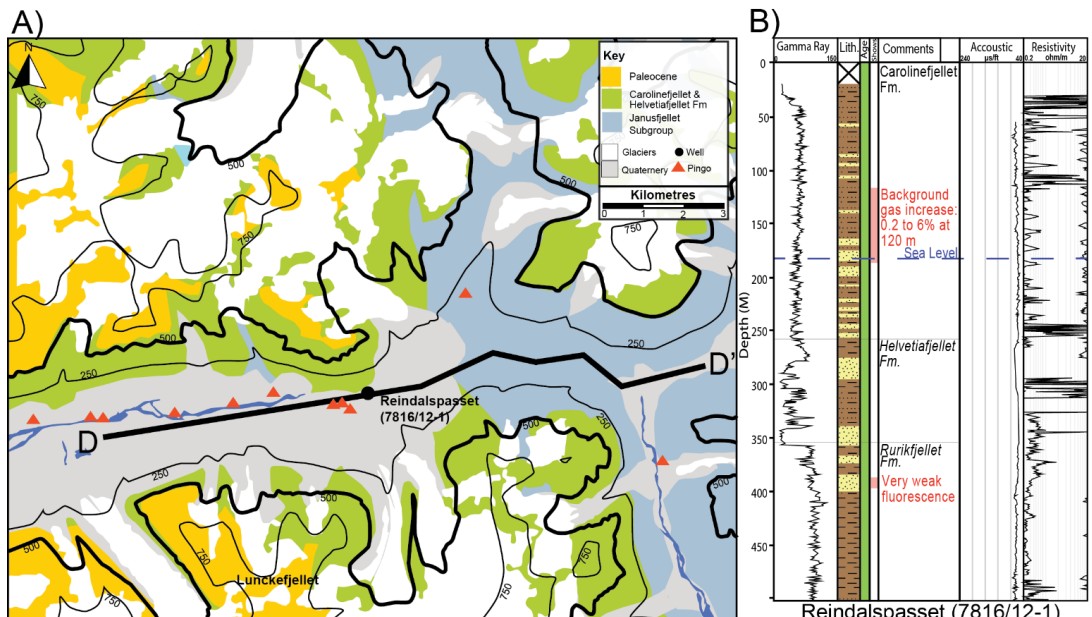

**Figure 12 – A) A Geological map of Reindalen ( base map data courtesy of © Norwegian Polar Institute) with the Reindalspasset borehole shown. Lunckefjellet plateau is also shown, where several coal boreholes that experienced fluid losses. The map location is shown in Fig. 1. B) The petrophysical log over shallow intervals in Reindalspasset (7816/12-1). The well sits in the valley of Reindalen where a series of pingos are situated updip from the wellbore, on the north side of the valley. Line D to D' shows the location of the seismic line shown in Fig. 17.**

No accumulations or gas influxes occurred in upper parts of this well though a background gas increase was observed. A 12 ¼" (31.115 cm) pilot hole was drilled to 164 m and background gas was recorded steadily at 0.2%. This hole was subsequently opened up to the planned 16" (40.64 cm); at 120 m depth background gas suddenly rose to 6% (Norsk Hydro, 1991). Because widening the hole resulted in greater fluid circulation, drillers and the wellsite geologist attributed the rise in gas due to thawing of the permafrost. They further speculated that it may be due to hydrate dissociation, though as no samples or pressures were measured this hypothesis remains impossible to assess. It is also important to note that this occurred in a low permeability siltstone interval where any gas accumulations are unlikely to flow at a good rate.

At Lunckefjellet, approximately 5 km southwest of the 7816/12-1 hydrocarbon well, permafrost has been demonstrated to be approximately 550 m thick (Juliussen et al., 2010). Drilling fluid losses were encountered in several boreholes on the plateau (Snsk, 2011b, c, d), well within the permafrost interval.

### 4.3.3 Edgeøya

The Plurdalen (7721/6-1) and Raddedalen (7722/3-1) wells are some 29 km apart (Fig. 13A) and were both drilled in 1972 by different operators. They both penetrate thick Permian carbonates and Carboniferous rift





successions. The uppermost 130 m of the Plurdalen well also encounters the lowermost parts of the Triassic
Botneheia shales.

The permafrost base here occurs in very hard, low porosity and low permeability rocks. Because of this
petrophysical data are, again, somewhat ambiguous here. Indeed, at Raddedalen the hard rocks in combination
with permafrost meant the well initially failed to make any progress during spudding (Total Marine Norsk,
1972). Drilling the upper hundred metres was very slow and wellbore cavings were also common, possibly due
to permafrost thawing. The drilling report for Raddedalen suggested the permafrost base was at 95 m (Total
Marine Norsk, 1972). They based this on resistivity peaks above 5000 Ωm at depths shallower than 95 m, but not
deeper, though we also note that this resistivity drop also coincided with a lithological change to shale. The
report also describes cycling and skipping in the acoustic log over this depth due to intermittent tool contact with
the wellbore wall caused by thawing at the permafrost base. The Raddedalen well data in Fig. 13B shows this but
also demonstrates that this skipping begins nearer 60 m depth. A water influx occurred at 224 m and probably
originated from the carbonate Wordiekammen Formation. The water influx was measured at 830 litres a minute
and had a temperature of 5°C, so clearly is from well below permafrost. Assuming the well-derived geothermal
gradient of 30°C/km (Betlem et al., 2018) this would put the base permafrost at 57 m depth, which matches well
with the observed skipping in the acoustic log. The aquifer is also overpressured by 4.41 bar and probably of
artesian origin.
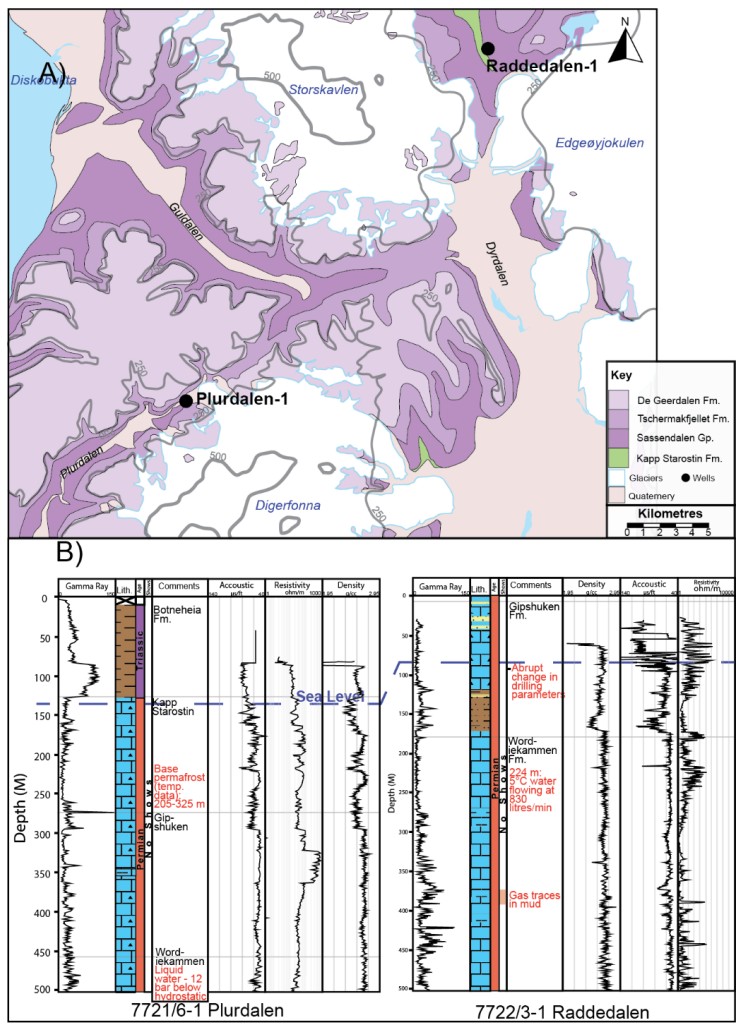

**Figure 13 - A) Geological map of central western Edgeøya (base map data courtesy of © Norwegian Polar Institute) showing the two hydrocarbon exploration wells on the island. The map location is shown in Fig. 1. B) Petrophysical logs from the hydrocarbon exploration wells at Plurdalen (7721/6-1) and Raddedalen (7722/3-1).**

At Plurdalen, a more complete set of petrophysical logs shows no clear evidence of permafrost or base permafrost. Temperature data apparently (Norske Fina a/S, 1972b, a) suggests base permafrost anywhere from 205 to 325 m. The log data does not appear to show any similar characteristics used to determine the base permafrost of the Raddedalen well. Liquid water, probably in the same Wordiekammen interval as at the Raddedalen well, was encountered at approximately 500 m. The temperature or salinity of this water was not recorded but, surprisingly, the pressure was 12 bar below hydrostatic. The most likely explanation for the underpressure is due to outflow and equilibrium with the fjord to the west, as 12 bar of underpressure at the wellhead corresponds to a hydraulic head approximately at sea level.





Neither wells encountered hydrocarbons, neither did the Plurdalen well report any shows. Raddedalen had minor
gas shows between 387-390 m and a trace increase in background gas in the mud returns (Norske Fina a/S,
1972a; Total Marine Norsk, 1972).

**4.3.4 Petuniabukta**

At Petuniabukta, Verba (2013) describes gas accumulations in Carboniferous reservoirs that do not have an
overlying lithological seal due to denudation and inclined and outcropping bedding. The author suggests a
permafrost interval of 250 to 400 m where no liquid water was encountered and suggests this must be sealing.
Oil has also been encountered in the area in small quantities (Senger et al., 2019). This indicates the
accumulations are likely thermogenic and that there are source rocks capable of generating and expelling
hydrocarbons, at least locally.

## 4.4 Gas Samples – Adventdalen, Tromsøbreen and Hopen

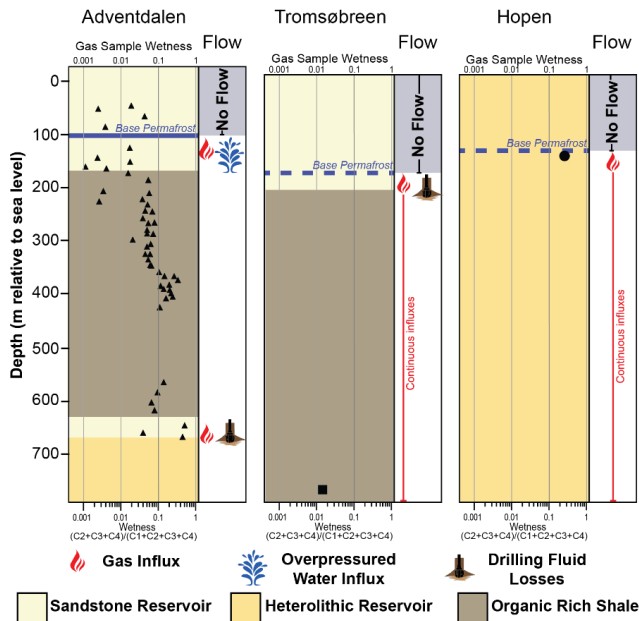

**Figure 14 – Gas wetness from samples taken in wells from Adventdalen (Ohm et al, 2019), Tromsøbreen (Norsk Polar**
**Navigasjon A/S, 1977b) and Hopen (Norske Fina A/S, 1972). The Hopen gas is much heavier and thus more prone to**
**form hydrates at lower pressures and higher temperatures than methane.**
Gas samples were taken by hydrocarbon exploration wells at Tromsøbreen and Hopen, with three samples taken
at each. Their compositions are shown in Table 4. A comprehensive analysis of the gas at the Longyearbyen $CO_2$
was carried out by Ohm et al. (2019) and Huq et al. (2017), and a rudimentary analysis of the coal borehole gas
discovery was carried out by Snsk (1981). These analyses of the gas accumulation and analysis of seeps of the



pingo systems in the area (Hodson et al., 2019) show the base-permafrost accumulation is methane dominated.
The more extensive analysis of the Longyearbyen $CO_2$ Lab gas provides a more complex story throughout the
entire stratigraphy with contributions from biogenic and thermogenic sources (Ohm et al., 2019). The sub-
permafrost gas at Hopen is much wetter and clearly from a thermogenic origin. Analysis at Tromsøbreen was
taken from gas much deeper than that encountered at base permafrost. However, it shows this gas is relatively
dry although still likely to be thermogenic due to the sample depth of 768 m and its extraction directly from the
Agardhfjellet Formation source rock (Norsk Polar Navigasjon a/S, 1977b, a). Figure 14 shows the wetness of gas
from the three locations, wetter gas means it has a greater component of heavier hydrocarbon molecules such as
ethane or propane.

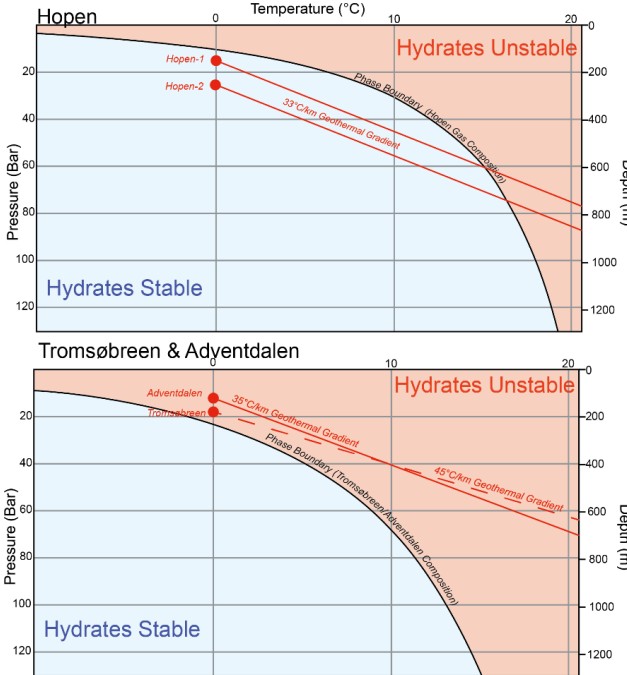


**Figure 15 – The upper graph shows the natural gas hydrate stability diagrams for the Hopen gas composition (Table**

**4). The lower graph shows the same for the Adventdalen (well DH4) and Tromsøbreen methane dominated gas**

**compositions (Table 4 and Fig. 14). Red circles represent the base permafrost (0° C) depths using pressures based on a**

**hydrostatic gradient from the surface. Lines represent stability with increasing depths at each locality based on local**

**geothermal gradients (Betlem et al., 2018; Isaksen et al., 2000).**

The composition of the gas is important in understanding its potential phase in the subsurface. Figure 15 shows
phase diagrams for the gas compositions and thermobaric conditions at Hopen, and at Adventdalen and
Tromsøbreen. While the dry gas at Tromsøbreen and Adventdalen is unlikely to be in hydrate form at their points
of discovery, the gas at Hopen is much wetter. As a consequence, it is more susceptible to be thermodynamically
stable as gas hydrate form (Betlem et al., 2019). In light of this, we modelled the potential gas hydrate stability





zone over Hopen based on the sampled composition. Figure 9C shows a thick zone where natural gas hydrates of
this composition are likely stable.

| Sample Number | Sample run 1 | Sample run 2 | Sample run 3 | Sample run 1 | Sample run 2 | Sample run 3 |
|---|---|---|---|---|---|---|
| | | | | | | |
| **Hyrocarbons** | **7617/7-1 (Tromsøbreen I)** | | | **7625/7-1 (Hopen I)** | | |
| **C1** | 64.79 - 70.81 | 68.57 | 63.84 | 92.35 | 94.97 | 97.24 |
| **C2** | 20.23 - 18.67 | 18.20 | 20.21 | 0.11 | 0.05 | 0.49 |
| **C3** | 10.97 - 7.76 | 9.26 | 11.10 | 0.09 | 0.01 | 0.16 |
| **C4** | 3.51 - 2.46 | 3.39 | 4.08 | 0.18 | 0.06 | 0.20 |
| **C5+** | 0.58 – 1.32 | 1.22 | 0.79 | 0.97 | 1.03 | 0.96 |
| **Nitrogen** | Abnormally High (not quantified) | | | 6.26 | 3.86 | 0.91 |
| **CO2** | - | | | 0.04 | 0.02 | 0.04 |
| **Gravity** | - | | | 0.609 | 0.600 | 0.591 |

**Table 4 – Geochemical data from samples taken at the hydrocarbon exploration wells at Tromsøbreen-1 and Hopen-1.**

## 5 Discussion

### 5.1 Identifying base permafrost

The active layer and upper parts of the permafrost interval are well-studied in Svalbard (Westermann et al., 2010;
Rachlewicz and Szczuciński, 2008; Strand et al., in press). However, the base permafrost is rarely the focus of
study with data coming overwhelmingly from industrial boreholes. Petrophysical data from predominantly
hydrocarbon wells may show some fluid trends that can be attributed to the transition from ice-bearing to water-
bearing strata. However, the complex geology largely overprints fluid responses. This is most likely due to the
generally low porosity of the rocks due to overcompaction due to deep burial and subsequent uplift.
Additionally, it may be reflective of the diffuse nature of the base permafrost. The most robust cases
demonstrating the base permafrost actually occur where there is very little change in the petrophysical data.
Because geology, rather than fluid content dominates the petrophysical response, the clearest cases are where
there are sudden fluid influxes into the wellbore with no change in the geological properties of the reservoir rock
itself. These influxes with no apparent lithological top seal occur in numerous locations throughout Svalbard,
most notably in multiple wells in Adventdalen, Hopen, Tromsøbreen, Gipsdalen, and Petuniabukta. These
occurrences show no particular prevalence with respect to age or depositional setting of the reservoir.
Permafrost is typically not considered to be present in coastal areas of Svalbard. However, evidence from
Tromsøbreen, and possibly also at Hopen, Petuniabukta and Kapp Laila, suggest that ice-bearing permafrost is
present in these areas (Fig. 3) and may even continue offshore.
The areas where permafrost has been modelled shows broad agreement with well-based observations in the
areas. Discrepancies are due to both the fact the modelled permafrost is based on temperatures, as per definition,
while well-based observations identify the base of ice-bearing permafrost which is also dependent on water
content, flow and its salinity. In addition, subsurface complexities are not captured in the model, for example the
forty-metre discrepancy between modelled and observed permafrost at Tromsøbreen is probably additionally





influenced by an overestimation of the geothermal gradient, the complex local geology and is in a heavily
glaciated area.

## 5.2 Permafrost formation and sealing effectiveness

Theoretically, the permafrost interval should form an extremely effective seal or "cryogenic cap". If ice-bound it
is impermeable, often thick and has the ability to self-seal through freezing water in the event of fracturing. In
reality the story is a little more complex and the seal-forming process is extremely poorly understood. An
effective permafrost seal is demonstrable in various locations in Svalbard by the presence of gas and abnormally
pressured water at the base of permafrost. This appears to be the case where the permafrost zone is ice-saturated,
most notably in valley settings. The previously described drilling losses in wells on the plateaus of Platåberget,
Breinosa Operafjellet, Lunckefjellet in addition to Ispallen (Snsk, 2014, 2013a, b, c) occurred in known
permafrost intervals (Juliussen et al., 2010). This suggests that in these highland areas, at least, that permafrost is
not forming a continuous impermeable seal.

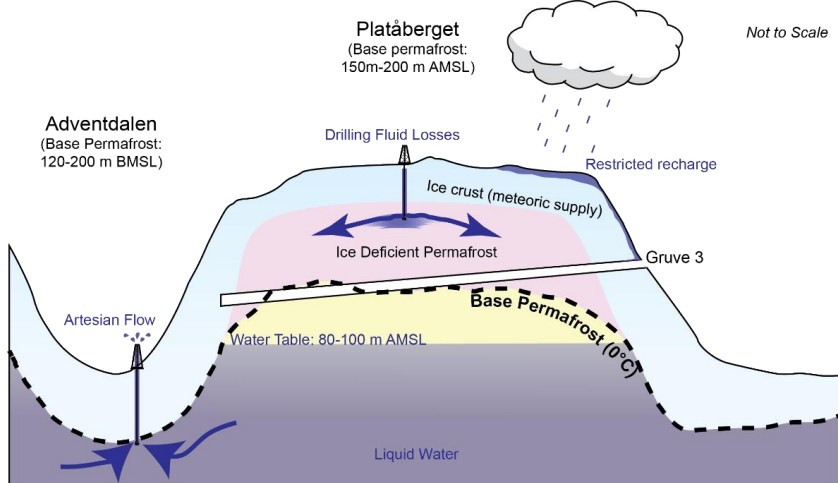


**Figure 16 – Schematic cross section of the permafrost interval at Platåberget. Artesian pressures in the valley wellbore**
**1967-1 suggest an elevated water table (Table 5) which still sits below the base of permafrost in the mountain. The**
**lack of water supply from below during permafrost formation leads to a dry and permeable permafrost interval and**
**subsequent drilling losses. Similar drilling fluid losses appear common in the permafrost interval in several plateau**
**areas in Svalbard. In the valleys the permafrost interval forms through the water table and results in a thick**
**impermeable ice seal.**
The valley-based permafrost interval has been shown to contain a proportion of liquid water (Keating et al.,
2018) in the form of microfilms and hypersaline pockets. Despite this, the interval remains impermeable to both
water and gas. Gas accumulations beneath the permafrost appear to be common and widespread regardless of
stratigraphy (Figs. 2, 3 and Table 3) which demonstrates the good sealing potential. Abnormal pressures are
common at the base of permafrost in several locations in Svalbard which demonstrates the sealing properties of
the overlying permafrost. The best data is in Adventdalen where sudden, slightly saline, water influxes occur at





the base of permafrost in the Helvetiafjellet Formation. The strong and sustained flow rates indicate appreciable
lateral connectivity within the aquifer, indicating an artesian origin of overpressure. The current view of this
overpressure is attributed to the formation of permafrost (Hornum et al., 2020) but the high flow rates
(Magnabosco et al., 2014),  reservoir connectivity and its outcropping beneath the fjord to the west (Blinova et
al., 2012) discount this.

| Case | High | Low |
|---|---|---|
| **Contact** | 160 m | 210 m |
| **Buoyancy Pressure (gas SG = 0.5537)** | 7.1 bar | 9.3 bar |
| **Aquifer Overpressure** | 6.9 bar | 4.7 bar |
| **Hydraulic Head Elevation (well: 32.5 m)** | 103 m AMSL | 79.2 m AMSL |

**Table 5 – Aquifer pressure calculation from wellhead pressures in well 1967-1 and the possible gas-water contact**
**wellbore 1971-10. The low case uses a saline water pressure gradient of 0.10067 bar/m while high case uses freshwater**
**gradient of 0.09795 bar/m.**

In highland areas the role of permafrost as a seal is less clear. Gas blowouts, like the documented occurrence in
well 1990-12 on Slaknosa plateau, were quite common based on anecdotal evidence. In the case of Slaknosa it is
likely that the permafrost acts as a seal. This is because the formations outcrop in the cliff sides so must require a
seal strong enough to withhold significant buoyancy pressure both above and laterally in the reservoir. However,
in other highland areas, including on Platåberget and Breinosa on the southern side of Adventdalen, the
permafrost interval appears to not be fully ice saturated.
The difference between the permafrost sealing potential in highlands and valleys can be explained by the
availability of water and the permafrost formation mechanism. Permafrost forms from the top-down, and as it
forms near the surface, it restricts the amount of meteoric input from the surface. As the permafrost thickens, a
water deficiency will develop if the water table remains deeper than the base of permafrost. Present day
pressures in Adventdalen (Table 5) suggest a hydraulic head well below the base permafrost which may explain
the water deficiency within the permafrost interval. This may lead to a thinner permafrost seal with potential
migration pathways through it, which may explain perennial springs at elevations up to 350 m around Breinosa.
In valley settings, the permafrost develops below the water table so there is always plentiful access to water,
resulting in a thick ice-saturated interval. This difference in water-availability during permafrost formation may
be critical to the development of an effective permafrost seal and explains why highland wells, such as those on
Platåberget, suffer drilling losses whilst those in the valley do not (Fig. 16). At Slaknosa, which is a highland
setting, the permafrost likely developed while having a constant water flux from the (presently) warm-based
glacier, Slakbreen, which is juxtapose and above the Slaknosa plateau. Regardless, the role of permafrost as a
seal in highland areas is clearly more complex than in the valleys. Another mechanism that could prevent water
from entering and freezing in the permafrost interval could be the early emplacement and trapping of gas.
Natural pathways through the cryospheric cap, even in areas of thick permafrost, are present in the form of
pingos, springs, warm-based glaciers, and beneath the fjords. Ice maybe more prone to fracturing, particularly in
shallow intervals where it is under little compression (Schulson, 2001). This may lead to fracture pathways





through the cryospheric cap although they likely self-heal through freezing water. At the Reindalen petroleum
exploration borehole pingos are situated up-dip and probably represent a natural leak point for gas (Fig. 17).
Elevated gas readings at 120 m in the wellbore likely represent a migration pathway at the base of permafrost
toward the pingos. Similarly, gas shows at Kapp Laila coincide with a potential migration pathway at the base of
permafrost. The crestal point of this carrier bed is a short distance offshore (Fig. 10) which is also coincidental
with the presence of pockmarks on the seafloor (Roy et al., 2015) where a potentially shallow permafrost tapers
out.

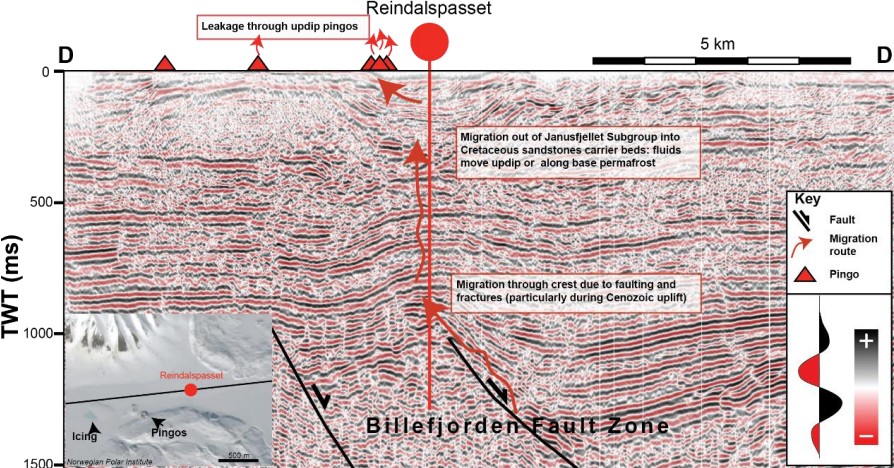


**Figure 17 – A seismic section intersecting the Reindalspasset wellbore, from Bælum and Braathen (2012). Deeper**
**thermogenic gas likely migrates through the crests of the Billefjorden Fault Zone. Further shallow migration occurs**
**through permeable Cretaceous stratigraphy and bypasses the permafrost seal through the pingo system to the west.**
**The location of the seismic section is shown in Fig. 12.**

## 5.3 Permafrost Traps

In order for gas to accumulate beneath the permafrost a trap must be present. The undulating base of permafrost
can form a trap and seal itself, or it may act as the top seal in combination with the underlying geology, these
examples are shown in Fig. 18. In the former, traps may form beneath mountains if the interval is water
saturated. This is because, although thicker, the base of permafrost is shallower than the surrounding valleys and
leads to natural concave-down structures for buoyant gas accumulations.  In valley settings where the base
permafrost forms a synclinal structure it is more likely that accumulations are situated within combination traps.
This is further supported by the fact the regional and local geology in Svalbard is rarely flat and contains
multiple lithological seals and reservoirs. In these traps a combination of structural geology, lithology and
permafrost properties contribute to developing hydrocarbon accumulations. This mechanism can be attributed to
the gas accumulation in Adventdalen (Fig. 19). The combination of this combination trap type and the ice-
saturated seals may explain why gas accumulations have been frequently encountered in valleys rather than





migrating and accumulating beneath shallower permafrost in highlands. Smaller accumulations, such as the one
encountered in Gipsdalen, may be restricted to localised undulations in the base-permafrost.

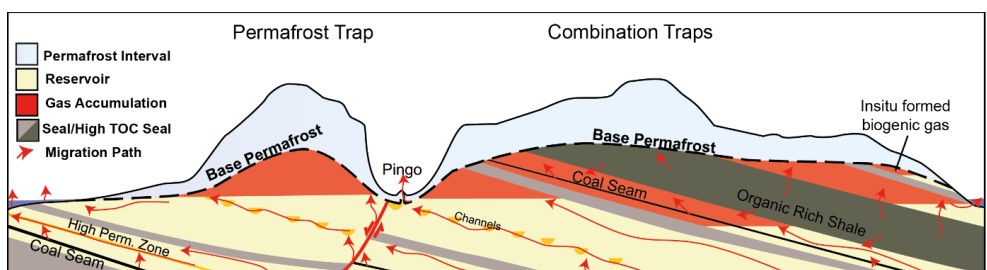

**Figure 18 – The different trapping mechanisms permafrost can provide. Undulations in the base permafrost alone**
**may form traps, which may be large under mountains if the permafrost seal is effective. Combination traps require**
**permafrost to contribute a lesser sealing surface area and appears to be the mechanism for trapping gas in**
**Adventdalen (Fig. 19).**
Gas trapped in hydrate form under the right thermobaric conditions is the exception to the previously discussed
trapping mechanisms. The gas sampled at Hopen is a strong candidate to originate from hydrates. The heavier
gas composition (Table 4) means it has a greater propensity to form hydrates at a given depth and temperature
(Fig. 9C). If the permafrost zone is drier in mountainous areas then it will mean the hydrostatic pressures beneath
them are lower than presently assumed. Therefore, they may be slightly less favourable for the formation of
natural gas hydrates due to lower-than-expected pressures.

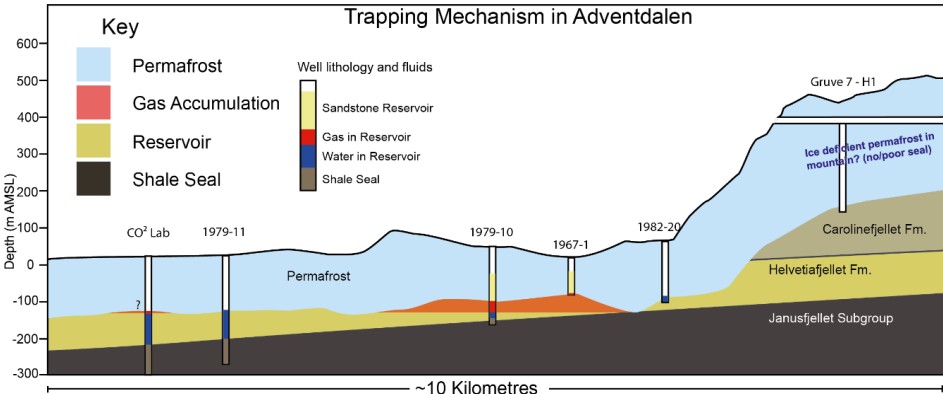

**Figure 19 – The potential combination-trapping style of the Adventdalen gas accumulation based on well**
**observations.**

## 5.4 Origins of gas

Gas originating from permafrost is typically attributed to a biogenic origin, primarily because thermogenic gas is
generated and migrates on much longer timescales. While biogenic gas is undoubtedly a contributor to sub-
permafrost gas in Svalbard (Hodson et al., 2019), thermogenic gas is also clearly a major contributor in several





locations (Ohm et al., 2019). In light of this, the lack of any accumulations or significant shows in the wells on
Edgeøya is probably due to the lack of any underlying prolific source rock.

Approximately 60% of wells in the Barents Shelf offer hydrocarbon shows (Senger et al., 2020), indicating that
the basin has at one point in the past been almost saturated with hydrocarbons. The large ultra-shallow
discoveries like Wisting, containing relatively unbiodegraded oil, are evidence of more geologically recent
hydrocarbon migration. This recent migration is almost certainly driven by major recent uplift over the past
thousands to hundreds-of-thousands of years (Henriksen et al., 2011).

Svalbard itself has undergone the greatest uplift of anywhere in the region, hence its existence as an archipelago.
The numerous prolific source rocks mean Svalbard is unique from other Arctic areas. Recent uplift has enabled
gas to escape directly from the source rocks or from deeper accumulations. The formation of permafrost has
effectively added a last line of defence preventing this gas from escaping to the atmosphere.

## 5.5 Timing and migration

Figure 20 is a petroleum systems chart with a focus on sub-permafrost accumulations in Svalbard. Clearly, all
other elements of the petroleum system must be present prior to migration taking place. Here the timescales are
binary with source, reservoirs and lithological seals forming tens or hundreds of millions of years ago and
permafrost forming during the past few or tens of thousands of years. Therefore, the most critical elements in this
system are the permafrost seal and gas migration.

The thermogenic gas must have been originated generated long before the formation of permafrost in the area
because the source rocks of the area are no longer deep enough to generate hydrocarbons. Recent migration is
almost certainly occurred during recent and ongoing uplift (Henriksen et al., 2011) due to repeated cycles of
glacial loading and unloading (Ohm et al., 2008). This has been ongoing throughout the Pleistocene and predates
permafrost formation. Therefore, the critical moment for most sub-permafrost gas accumulations in Svalbard is
the timing of permafrost formation itself. The exception to this is in the extremely young moraine sediments in
the case of Kapp Amsterdam, which also highlights ongoing gas migration.

Gas migration will occur through permeable intervals, typically at the crest of structures. Faults may aid the
movement of gas from deeper structures, particularly during uplift and fault reactivation as appears to be the case
at Reindalen (Fig. 17) which sits on the Billefjorden Fault Zone (Bælum and Braathen, 2012). The discovery of
shale gas in Adventdalen (Ohm et al., 2019) also shows that source rocks still internally trap large amounts of
gas. This gas will have migrated directly out of source rocks during uplift due to gas expansion and rock
fracturing.



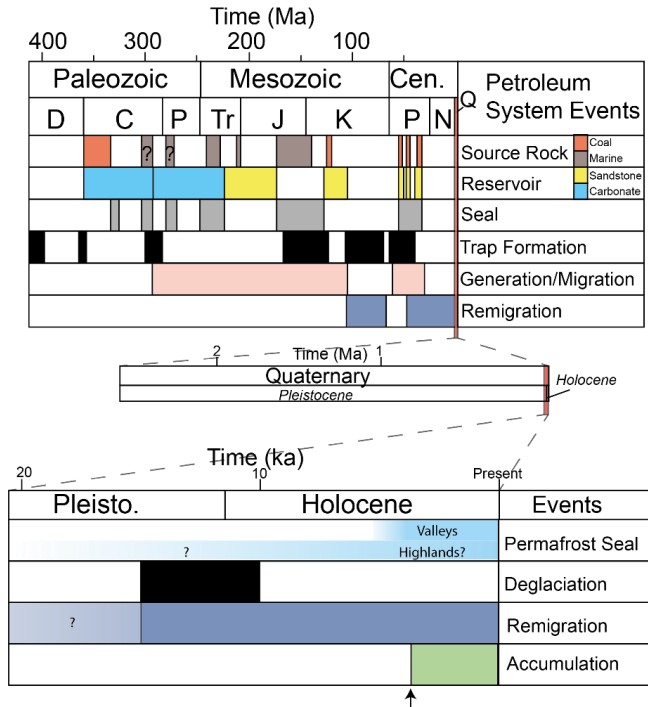

**Figure 20 - Petroleum systems chart for Svalbard. The upper part covers the important elements of the past 400 million years whereas the lower parts show the importance of the most recent events. The critical moment is the timing of permafrost formation, which is also evidence that gas migration must also have been occurring recently, and most likely ongoing today.**

## 5.6 Size, frequency and consequences of gas accumulations

Gas accumulations beneath permafrost appear to be a common occurrence in Svalbard and they show no preference to stratigraphic age or geological setting. It is important to remember that none of the wells that encountered sub-permafrost gas were actually looking for it, indeed most hydrocarbon exploration wells aim to avoid such shallow gas accumulations (Ronen et al., 2012).In this study, of eighteen hydrocarbon wells in Svalbard, eight show good evidence of permafrost (44%). Four of these permafrost bearing wells show moveable gas accumulations at the base of permafrost (22% of all wells or 50% of permafrost bearing wells), three clearly show no presence of an accumulation while one contains gas shows. Expanding this to all wells in this study, 18 show evidence of permafrost and 9 of these showing evidence of gas accumulations (50%), though the coal wells for this study were obviously selected in areas of interest. This is an extremely high success rate for something that was not being looked for, and thus highlights the likelihood that these gas accumulations are a very common occurrence. For reference, the Barents Shelf has one of the highest technical success rates in the world at just below 50% (Norwegian Petroleum Directorate, 2020) for prospects that have been specifically targeted using advanced geological and geophysical methods.



As with conventional hydrocarbon accumulations, the size of sub-permafrost accumulations probably varies
significantly. The accumulation in Adventdalen is relatively significant, but also of little economic interest; the
1967-1 well produced in excess of 2.5 million cubic metres of gas between 1967 and 1975 (Snsk, 1981). Despite
being of little economic interest, these accumulations may still provide an alternative and cleaner energy source
than coal, which is presently used to generate power in Svalbard. Unfortunately the data are quite poor because
the well was also periodically shut in over this time. Speculatively, if the convex-up shaped base permafrost
below mountains acts as an effective trap then volumes may be even larger than the (relatively) better understood
accumulations, in the valleys. Given the encountered overpressures in both water and gas bearing rocks it is fair
to assume that the permafrost seal can withstand significant buoyancy pressures or large gas columns. It is more
likely that the accumulations are regulated laterally by natural pathways through the permafrost at pingos, fjords
or glaciers.

| Area of Hopen | 46.12 km$^2$ |
|---|---|
| Approximate thickness of hydrate stability zone | 600 m (This study) |
| Net to Gross (sandstone) | 25% (Hynne, 2010) |
| Average Porosity | 14% (Mørk, 2013) |
| Volume as free gas | 968 Million Sm$^3$ |
| If Hydrate | 154.963 Billion Cu. m. |

**Table 6 – Estimation of gas volume under Hopen using properties from the stated publications. This assumes the**
**stability zone is saturated to its base, which is highly dependent on the migration rate of gas. This may be somewhat**
**unreasonable to assume but it is worth noting that the wells did monitor persistent gas influxes throughout the entire**
**interval.**
Because the sub-permafrost accumulations are relatively shallow and under lower pressure, the gas will be much
less dense, and thus voluminous, than conventional deeper accumulations. The exception to this is if the gas is in
hydrate form where methane concentrations are some 160 times higher than in free gas form (Majorowicz and
Hannigan, 2000). Table 6 shows the potential volumes of gas within the hydrate stability zone beneath Hopen
using typical net to gross and reservoir properties for the De Geerdalen Formation (Mørk, 2013; Hynne, 2010).
The calculations show volumes for both free gas, under atmospheric pressure and if it is in hydrate form.
Given the sparse data and bias in drilling locations it is impossible to be very quantitative with respect to the size
and frequency of these accumulations. What is evident is that permafrost is acting as an ultimate seal to these
accumulations, and that they are numerous, and, based on the only occurrence where flow was recorded, on the
orders of million cubic metres.

## 5.7 Regional distribution

Based on the occurrences in Svalbard, the prerequisites for sub-permafrost gas to accumulate are, firstly, an
impermeable (ice-saturated) permafrost layer, secondly, a source of gas and, finally, gas migration at a time after
permafrost formation. Much of the Circum-Arctic shares a similar geological history with Svalbard. A major
source of migrating gas in Svalbard is likely from the Mesozoic source rocks (Ohm et al., 2019), which can also



be found in the Russian and North American Arctic (Leith et al., 1993; Polyakova, 2015). Recent uplift caused
by isostatic rebound has left fluids in the subsurface on the Barents and Svalbard out of pressure equilibrium and
driving present-day migration (Birchall et al., 2020). Svalbard shares its Pleistocene glacial history with the
Circum-Arctic (Batchelor et al., 2019) so it is not unreasonable to expect sub-permafrost gas accumulations to be
regionally widespread. Indeed, gas emanating from zones of permafrost is well-documented onshore and
offshore in the Russian Arctic, particularly in hydrocarbon provinces (Chuvilin et al., 2020 and references
therein) and as natural gas hydrates (Yakushev & Chuvilin, 2000).

## 6 Conclusion

Although gas at the base of permafrost has been encountered frequently during more than fifty years of drilling
in Svalbard, it has not been studied or widely recognised until now. In this study we have provided a synthesis of
historical and modern observations and their implications. Our key findings are:
• Gas accumulations trapped at the base of permafrost occur throughout the archipelago in several
stratigraphic intervals.
• The gas accumulations are evidence for ongoing hydrocarbon migration
• Gas encountered in wellbores on Hopen is compositionally heavier and likely within the gas hydrate
stability zone
• Permafrost is a good seal in valleys but appears to possess permeable intervals in highland areas
• Groundwater flow below permafrost is much greater than previously documented
• There is evidence of relatively thick coastal permafrost, particularly in eastern Svalbard
Because methane is a potent greenhouse gas and the Arctic is warming faster than anywhere else on Earth (Lind
et al., 2018), the release of sub-permafrost gas accumulations in Svalbard may contribute a positive climatic
feedback effect. Shallow gas associated with permafrost has been documented throughout much of the Circum-
Arctic (Nielsen et al., 2014; Minshull et al., 2020; Hodson et al., 2020; Chuvilin et al., 2020). Because Svalbard
shares much of its geological and glacial history with the Circum-Arctic it seems likely that the gas
accumulations we document in Svalbard are more widespread.

## Acknowledgements

This research is funded by the Research Centre for Arctic Petroleum Exploration (ARCEx) partners and the
Research Council of Norway (grant number 228107), CLIMAGAS (grant number 284764) and the Norwegian



CCS Research Centre (grant number 257579). We sincerely appreciate access to Store Norske Spitsbergen
Kulkompani's vast internal archives and to the data from the Longyearbyen CO2 lab project (http://co2-
ccs.unis.no). We also thank Sarah Strand for fruitful discussions on permafrost dynamics.



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



## Author Contribution

**Thomas Birchall:** Conceptualisation, methodology, validation, investigation, data curation, writing – original draft, writing – review and editing, visualisation, project administration.

**Malte Jochmann:** Conceptualization, Validation, Investigation, Resources, Data Curation, writing – reviewing

**Peter Betlem:** Methodology, Software, Validation, writing – reviewing

**Kim Senger:** Conceptualization, methodology, validation, resources, writing – reviewing, supervision

**Andrew Hodson:** Validation, writing – reviewing, supervision

**Snorre Olaussen:** Validation, writing – reviewing, supervision

## Competing Interests

The authors have no known competing interests.

## Data Availability

The historical nature of the data and reports means they are available in hard-copy only. Reports referenced in this article are proprietary to their respective companies.

For permafrost and hydrate stability modelling herein, the methodology is detailed in the following publication: https://doi.org/10.1016/j.marpetgeo.2018.10.050