# Peer review of "Review Article: Permafrost Trapped Natural Gas in Svalbard, Norway"

_The Cryosphere, 2021_

## Referee Comment (RC2)

[referee-annotated manuscript omitted]

---

## Referee Comment (RC3)

**1 Permafrost Trapped Natural Gas in Svalbard, Norway**

Authors: Thomas Birchall*[1, 2], Malte Jochmann[1, 3], Peter Betlem[1, 2], Kim Senger[1], Andrew

Hodson[1], Snorre Olaussen[1]

[1]Department of Arctic Geology, The University Centre in Svalbard, P.O. Box 156, N-9171 Longyearbyen,

Svalbard, Norway

[2]Department of Geosciences, University of Oslo, P.O. Box 1047, Blindern, 0316 Oslo, Norway

[3]Store Norske Spitsbergen Kulkompani AS, Vei 610 2, 9170 Longyearbyen, Svalbard. Norway

*Correspondence to: Thomas Birchall (Thomas.birchall@unis.no)

**Abstract.** Permafrost has become an increasingly important subject in the High Arctic archipelago of Svalbard.

However, whilst the uppermost permafrost intervals have been well studied, the processes at its base and the impacts of the underlying geology have been largely overlooked. More than a century of coal, hydrocarbon and scientific drilling through the permafrost interval shows that accumulations of natural gas trapped at the base permafrost is alarmingly common. They exist throughout Svalbard in several stratigraphic intervals and show both thermogenic and biogenic origins. These accumulations combined with the relatively young permafrost age indicate gas migration, driven by isostatic rebound, is presently ongoing throughout Svalbard. The accumulation sizes are uncertain, but one case demonstrably produced several million cubic metres of gas over eight years. Gas encountered in two boreholes on the island of Hopen appears to be situated in the gas hydrate stability zone and thusly extremely voluminous. While permafrost is demonstrably ice-saturated and acting as seal to gas in lowland areas, in the highlands it appears to be more complex, and often dry and permeable. Svalbard shares a similar geological and glacial history with much of the Circum-Arctic meaning that sub-permafrost gas accumulations are regionally common. With permafrost thawing in arctic regions, there is a risk that the impacts of releasing of sub-permafrost trapped methane is largely overlooked when assessing positive climatic feedback effects.

**26 Keywords**

Permafrost; Top seal; Natural Gas; Cryosphere; Greenhouse Gas; Arctic; Greenhouse Gas; Hydrates.

**29 1 Introduction**

I don't think using the term 'generally accepted' is reasonable if you can only cite one reference- please add more

It is generally accepted that thawing permafrost results in the release of methane gas to the atmosphere (Knoblauch et al., 2018). Methane is a potent greenhouse gas and its release from permafrost acts as a positive climatic feedback loop (Boucher et al., 2009; Howarth et al., 2011; Lashof and Ahuja, 1990). The Arctic is

*explain to the reader why pf is more sensitive due to the W. Spitsbergen current?- warmer ground temps?, more variable climate… explain*

particularly sensitive to climatic changes and Svalbard is even more so due to the West Spitsbergen Current (Divine and Dick, 2006; Van Pelt et al., 2016; Aagaard et al., 1987). Svalbard is, therefore, a critical site for studying the evolution of permafrost and sub-permafrost processes (Hornum et al., 2020; Hodson et al., 2019;

Christiansen et al., 2010; Isaksen et al., 2000).

While methane emissions from thawing of the permafrost active layer is relatively well understood (Knoblauch et al., 2018; Vonk and Gustafsson, 2013), the prevalence and volumes of gas accumulations trapped beneath the permafrost "cryospheric cap" (Anthony et al., 2012) has been much less studied. Here we present evidence of

*accumulation no… occurrences perhaps*

such gas accumulations  in Svalbard, where the relatively young permafrost (Gilbert et al., 2018) appears to be

*you have not presented evidence of signficant accumulations only occurrence*

regionally sealing significant gas accumulations.  The gas here may originate from biogenic or thermogenic processes (Hodson et al., 2019; Ohm et al., 2019) and may  be in free-form or, under the right compositional and thermobaric conditions, in the form of natural gas hydrates (Sloan Jr et al., 2007; Betlem et al., 2019).

*generally most hydrate researchers simply use term pressure temperature conditions… key point however is that PT conditions do not dictate occurrence of gas in hydrate form.  Need to also consider pore water salinity, porous media setting access to pore water and gas*

Occurrences of gas originating from within or below intervals of permafrost are typically identified in studies on natural gas hydrates and have been documented in both the Russian (Chuvilin et al., 2000; Makogon and

Omelchenko, 2013; Yakushev and Chuvilin, 2000; Skorobogatov et al., 1998; Chuvilin et al., 2020) and North

American Arctic (Bily and Dick, 1974; Collett et al., 2011; Kamath et al., 1987; Majorowicz and Hannigan,

*You are missing many important Mackenzie Delta references… But key point that you should keep in mind is that many if not all of N. A. hydrate occurrences in these references are well beneath the base of permafrost. They are conditioned not by permeability variability at the base of permafrost, but by the PT environment set up by thick pf occurrences.*

2000; Nielsen et al., 2014).

Permafrost is defined as ground that remains at sub-zero (in Celsius) temperature for more than two consecutive years, regardless of fluid content. Physically speaking, ice-saturated permafrost possesses extremely good sealing

*Keating et al seems to not make any mention of sealing properties-you are being declarative 'extremely good' but not provided justification …*

[revised manuscript text omitted]

what sort of challenge… unexpected hole erosion? variable geopressure? fluid loss

Permafrost often poses a challenge to geologists, particularly for drilling boreholes (Vrielink et al., 2008), and acquiring and processing seismic data (Schmitt et al., 2005; Johansen et al., 2003). This is because it changes the properties of shallow unlithified sediments to become much more rigid and cemented by ice. Therefore, the permafrost interval has much faster seismic velocities and can lose mechanical competence as it is drilled through with heated or saline fluids. The near-surface rocks in Svalbard are typically well cemented and very rigid due to deep burial and subsequent uplift.

Many classic pf reference you cite are for unconsolidated sediments.. your setting is v. different - this points to a missing discussion on petrophysics…  what type of cement, what ranges in porosity and permeability, etc. etc.  and how are your 'well cemented and very rigid' bedrock occurrences expected to behave at negative temperatures- the readers of this journal will want to know this.

[Figure]

This figure is v. weak;
-text is  hard to decipher
-numbering is almost impossible to decipher
-legend for bedrock cites age, but for this paper the reader is interested in lithology
-

[revised manuscript text omitted]

**3 Data and methods**

Several decades of coal and petroleum exploration, as well as research drilling, in Svalbard has led to much anecdotal evidence of gas accumulations beneath permafrost. We have attempted to verify this by analysing data from boreholes that have penetrated through the permafrost in Svalbard. These boreholes include eighteen hydrocarbon exploration wells, ten scientific boreholes, eight of which are from the Longyearbyen $CO_2$ Lab (from two drill sites). Also integral to this study are the somewhat sporadic data, including drilling and geological reports, from more than five hundred coal exploration boreholes drilled by the local Store Norske Spitsbergen Kulkompani (SNSK) over a period of nearly a century. We identify where gas accumulations occur and where these coincide with the base of permafrost, or the first permeable interval below it. One of the major challenges with these boreholes is that they typically target much deeper stratigraphy and often acquire very limited petrophysical data in the shallow parts. Typically, only the gamma ray logging tool, which measures the rocks natural radioactivity, is run in the shallow intervals. The available well data used in this study are presented in Table 1. Ascertaining the presence of sub-permafrost gas presents several challenges.

Identifying the presence of permafrost is simple and can often be clear from geomorphological features such as pingos. However, identifying the thickness and base of permafrost is much more challenging (Osterkamp and Payne, 1981). Table 2 shows the ideal responses of petrophysical and drilling data at the lower permafrost boundary and the challenges to each method. By far the biggest challenge to petrophysical and drilling data analysis in Svalbard is due to the low porosity, heterolithic, very rigid and overcompacted rocks (Henriksen et al., 2011). The nature of the base of permafrost itself is also not well understood, but it is a reasonable assumption that it a diffuse boundary which adds to the complexity of identifying a permafrost boundary in petrophysical data alone. Further complications arise from the drilling fluid used and circulated in the wellbores which was often heated and hypersaline. Nevertheless, it is approximate base of permafrost on a case-by-case basis using all avail lithology and drilling data is useful in identifying change it is obvious evidence of being below the ice-bearing salinity). Other indicators that can help identify the position the wellbore, sudden changes in the character or amount of cutting returns and increases in background gas measurements. The strongest indication of base permafrost occurs where fluid influxes into or out of the wellbore suddenly occur in thick, normally permeable sandstones. In this situation it is very likely it is due to the transition of impermeable permafrost to permeable water or gas-bearing rock. Abnormally high pressures at the apparent base of permafrost are often mentioned in well reports and provide good evidence that the permafrost is acting as an effective seal.

I find pf interpretation with just variations water flux or in permeability to be v. weak. It would be much stronger if you had at least one other measurement to confirm

| | | Petrophysics - Start of Data (m MD) | | | | | | Gas Data | | |
|---|---|---|---|---|---|---|---|---|---|---|
| Well | Gamma Ray | Resistivity | Acoustic | Density | Temperature | Cuttings | Gas Shows (Chromatograph) | Fluid Samples | Well Report |
| Hydrocarbon Exploration | | | | | | | | | |
| 7617/7-1 Tromsøbreen-1 | (Drilling param | | | | | | | | Y |
| 7617/7-2 Tromsøbreen-2 | 17 | 350 | | | | | | | Y |
| 7625/7-1 Hopen-1 | 3.5 | - (SP logged) | - | - | BHT | Surface | Surface | c. 150 m | Y |
| 7625/6-2 Hopen-2 | Surface | 349 | 349 | 638 | Surface | Surface | Surface | - | Y |
| 7714/2-1 Grønnfjorden | not logged | | | | | Cored | - | - | Y |
| 7714/3-1 Bellsund | ? | | | | | | | | N |
| 7715/1-1 Vassdalen-2 | Surface | 17 | - | - | - | - | - | - | N |
| 7715/1-2 Vassdalen-3 | - | - | - | - | - | - | - | - | N |
| 7715/3-1 Ishøgda | Surface | Surface | Surface | Surface | Surface | Surface | - | - | N |
| 7721/6-1 Plurdalen | 5 | 83 | 5 | 83 | Surface | Surface | Surface | Water at 500 m | Y |
| 7722/3-1 Raddedalen | Surface | 5 | 591 | 593 | 5 | Surface | Surface | - | Y |
| 7811/2-1 Kvadehuken-1 | not logged | | | | | Cored | - | - | N |
| 7811/2-2 Kvadehuken-2 | not logged | | | | | Cored | - | - | N |
| - Kvadehuken-0 | Shallow, no data | | | | | | | | |
| 7811/5-1 Sarstangen | 30 | 615 | | | BHT | Surface | 260m | - | Y |
| 7814/12-1 Kapp Laila | Surface | - (SP logged) | | | | 24 m (partial recovery) | | | Y |
| 7815/10-1 Colesbukta | Surface | 41 | 1467 | - | - | - | - | - | N |
| 7816/12-1 Reindalspasset | From surface | 22 (induction) | 22 | 22 | 17.4 | Suface | 20 m | - | Y |
| Selected Coal Boreholes | | | | | | | | | |
| 1967-1 Adventdalen | | | | | | | Y | - | Y |
| 1979-10 Adventdalen | Cored (not logged) | | | | | | - | - | Y |
| 1979-11 Adventdalen | | | | | | | - | - | Y |

It is not possible to fully interpret this table…the formatting is sloppy and it is not clear what the numbers mean, the term surface, etc. ? please add interpreted base of permafrost for each well?

| | | | | | | | | | |
|---|---|---|---|---|---|---|---|---|---|
| DDH1B Gippsdalen | | | | | | | - | - | Y |
| 1982-20 | No data | | | | | | - | - | 1982 drilling summary |
| Gruve 7 - H1 | | | | | | | Y | - | 1979 drilling summary |
| 1981-02 | | | | | | | - | - | 1981 drilling summary |
| 1981 (Platåberget) | | | | | | | - | - | 1981 drilling summary |
| 1981-05 Breinosa | | | | | | | - | - | 1981 drilling summary |
| 1981-06 Breinosa | Cored (not logged) | | | | | | - | - | 1981 drilling summary |
| 1979-1 Reindalen | | | | | | | - | - | Y |
| 1990-12 Slaknosa | | | | | | | - | - | Y |
| Scientific Boreholes | | | | | | | | | |
| DH1 | 3 | 3 | 9 | - | Surface | Cored | - | | Y |
| DH2 | 10 | 10 | 10 | - | Surface | Cored | - | - | Y |
| DH3 | Cored: not logged | | | | | Cored | - | - | Y |
| DH4 | Surface | 440 | 440 | - | Surface | Cored | - | Throughout | Y |
| DH5r | 3 | - | 100 | - | Surface | Cored | - | Below 645 m | Y |
| DH6 | Cored: not logged | | | | | Cored | - | | Y |
| DH7a | Cored: not logged | | | | | Cored | - | Below 645 m | Y |
| DH8 (Shallow) | Cored: not logged | | | | | Cored | - | - | Y |
| BH10-2008 | Surface | 67 | 48 | Surface | - | - | - | - | Y |
| Janssonhaugen (temperature) | - | - | - | | Surface | - | - | - | N |

**Table 1 – Data availability and intervals recorded for the permafrost penetrating boreholes.**

Poorly worded.. thermometer.. or thermistors? direct measurement of mud temp or in situ measurement of ground temperature?

Direct temperature data from thermometers used in conjunction with wireline logging tools is common from hydrocarbon exploration wells. However these were rarely allowed to reach thermal equilibrium with the surrounding formations following drilling and fluid circulation. Therefore accurate absolute temperature measurements are rare, though temperature trends (e.g. inflection points) can be used more qualitatively to estimate base permafrost. Wells monitored over longer time periods, such as the scientific boreholes in

Adventdalen (Isaksen et al., 2000; Olaussen et al., 2019; Juliussen et al., 2010) are relatively rare, but provide much more reliable and precise temperature data.

Identifying the presence of gas is relatively simple and, although petrophysical data is generally not helpful in shallow sections for fluid discrimination. Reliable evidence comes from influxes of gas into the wellbore which has been sampled from wells in Adventdalen, Tromsøbreen and Hopen (Senger et al., 2019). Elevated background gas is another good indicator of sub-permafrost gas and is measured in drilling fluids returning to the surface and extracted by a "gas trap". This method typically identifies in-place dry gas accumulations or gas that has exsolved from fluid on its way to the surface due to pressure decline. However, these measurements do not detect gas that remains dissolved in formation water. Gas from drilling mud is also impacted by a variety of factors (Marum et al., 2019), including drilling rate, drilling mud type and, perhaps the most pertinent, temperature; low temperatures can cause heavier hydrocarbons to condense and avoid detection, it also causes drilling fluids to become more viscous, further inhibiting gas extraction.

*Well you just said gas exsolved due to pressure decline.. but this implies that this is gas which was dissolved in situ… why would the remaining dissolved gas be different*

[revised manuscript text omitted]

in Fig above you use the terms mobile gas and gas shows... here you use evidence for gas and tentative/shows. None of these terms are well described

| Well | Evidence for Gas Under Permafrost | Tentative/Shows | Permafrost but no gas |
|------|-----------------------------------|-----------------|-----------------------|
| *Hydrocarbon Exploration* | | | |
| **7617/7-1 Tromsøbreen-1** | • | | |
| **7617/7-2 Tromsøbreen-2** | • | | |
| **7625/7-1 Hopen I** | • | | |
| **7625/6-1 Hopen II** | • | | |

| Well | | | |
|---|---|---|---|
| **7714/2-1 Grønnfjorden** | | | |
| **7714/3-1 Bellsund** | | | |
| **7715/1-1 Vassdalen-2** | | | |
| **7715/1-2 Vassdalen-3** | | | |
| **7715/3-1 Ishøgda** | | | |
| **7721/6-1 Plurdalen** | | | • |
| **7722/3-1 Raddedalen** | | | • |
| **7811/2-1 Kvadehuken-1** | | | |
| **7811/2-2 Kvadehuken-2** | | | |
| **Kvadehuken-0** | | | |
| **7811/5-1 Sarstangen** | | | |
| **7814/12-1 Kapp Laila** | | • | |
| **7815/10-1 Colesbukta** | | | |
| **7816/12-1 Reindalspasset** | | | • |
| *Coal* | | | |
| **1967-1 Adventdalen** | • | | |
| **1979-10 Adventdalen** | • | | |
| **1979-11 Adventdalen** | | | • |
| **1982-20 Adventdalen** | | | • |
| **Gruve 7 - H1 Adventdalen** | | | • |
| **DDH1B Gippsdalen** | • | | |
| **1979-1 Reindalen** | | • | |
| **1981 Platåberget** | | | |
| **1981-Breinosa** | *TD above base permafrost* | | |
| **Lunckefjellet** | | | |
| **Ispallen** | | | |
| **1990-12 Slaknosa** | • | | |
| *Scientific Wellbores* | | | |
| **DH1** | | | |
| **DH2** | | | |
| **DH3** | | | |
| **DH4** | • | | |
| **DH5r** | • | | |
| **DH6** | | | |
| **DH7a** | | | |
| **DH8 (Shallow)** | | | |
| **BH10-2008** | | | • |
| **Janssonhaugen** | *TD above base permafrost* | | |

**Table 3 – Wells showing where gas is and is not present at the base of permafrost. Wells without a bullet either contain no permafrost or no relevant data.**

**4.2 Case Studies: confirmed sub-permafrost gas**

**4.2.1 Adventdalen**

Svalbard's largest settlement, Longyearbyen, is located in Adventdalen (Fig. 4), and one of the better studied areas of Svalbard (Hodson et al, 2020; Hornum et al, 2020; Johansen et al., 2003; Beka et al., 2017; Olaussen et al., 2019 and references therein). The wells of the Longyearbyen $CO_2$ Lab and coal exploration boreholes of SNSK both show the presence of gas beneath the permafrost in Adventdalen (Fig. 5) provides a correlation panel of these wellbores.

[Figure]

**Figure 4 - A Geological Map of Adventdalen showing some of the youngest stratigraphy exposed in Svalbard. The profile A to A' represents the well correlation in Fig. 5 and B to B' the modelled permafrost profile in Fig. 6.**

At the near-coast drillsite-1 of the Longyearbyen $CO_2$ Lab wells temperature data from DH1 and DH2 indicate a thin permafrost interval with the base at approximately 20-30 m (Beka et al., 2017). Although sub-zero temperatures were recorded at this site, the presence of ice is strongly dependent on the pore-fluid salinity. At drillsite-2, wellbores DH3 and DH4 encountered overpressured water at the base permafrost. DH4 and DH5R also encountered significant natural gas with this water kick and it was collected in gas bags for sampling (Ohm et al., 2019; Huq et al., 2017). Temperature logs from well DH4 suggest base permafrost from 150 to 200 m depth but given the drilling fluid losses and mud circulation there is considerable uncertainty in this data. Cores from nearby wells DH6 and DH7A also show elevated methane levels at this depth. The water and gas influxes occur somewhere towards the middle (i.e. not top of the reservoir) of the sandstone dominated Helvetiafjellet Formation. Figure 6 is the modelled permafrost thickness in Adventdalen which shows good agreement with the independent well data observations.

[Figure]

**Figure 5 – A Well correlation of base permafrost with all available geological, drilling and** section is shown in Fig. 4. The wells in this section

**highlight the somewhat sporadic nature of data availability over the shallow, permafros**

In the same area, hundreds of coal boreholes, drilled by SNSK over the decades, have penetrated the permafrost interval, although data for these is more fragmented. Well 1979-11 was drilled approximately two kilometres south of Longyearbyen $CO_2$ Lab drillsite-2 in Endalen. This well encountered water influxes with no mention of gas, although no depths are stated in the report (Snsk, 1980, 1981). Well 1979-10, two kilometres to the southeast in the neighbouring valley Todalen encountered methane-rich gas overlying inflowing water at the base of permafrost at a depth between 150 to 200 m (Snsk, 1981, 1982b; Leythaeuser et al., 1984). Well 1967-1, approximately three kilometres east and geologically updip of 1979-10, reached a depth of 106 m where a gas accumulation was encountered (Snsk, 1981).  This well was also the subject of considerable interest by SNSK

who investigated the potential of producing the gas commercially. Well 1982-20, approximately one kilometre southeast of 1967-1, at the base of Breinosa and the coal mine Gruve-7, did not encounter gas and took water influxes of 33-42 litres per minute at approximately 150 m at the base of permafrost (Snsk, 1982a). Another reported well, named only "first water well", (Snsk, 1982a) in the same area flowed from the same interval at 40-

50 litres per minute. Water from these two wells had a measured chloride concentration of 1500 ppm (Snsk,

1982a).  A well drilled inside Gruve-7 at approximately 380 m AMSL encountered liquid water at 154 m depth.

[Figure]

**Figure 6 - Modelled permafrost thickness** ⟨...⟩ **own in Fig. 4. The model**

**parameters are discussed in the metho** ⟨...⟩ **al is entirely based on temperature**

**rather than ice thickness or presence.**

*[Note annotation: What depth was this test conducted at and how does this fit with interpretations of the base of permafrost?.. I would delete this discussion]*

Well 1967-1 and 1979-10 most likely ⟨...⟩ while well 1982-20 encountered permafrost over the same stratigraphic ⟨...⟩ down-dip of the gas-water interface. Intermittent flow from the ⟨...⟩ ween October 1967 and July 1975 (Snsk,

[revised manuscript text omitted]

*Any discussion of hydrate thicknesses seems v. speculative to me and including hydrate volume absolutely speculative.*

**Table 6 – Estimation of gas volume under Hopen using properties from the stated publications. This assumes the stability zone is saturated to its base, which is highly dependent on the migration rate of gas. This may be somewhat unreasonable to assume but it is worth noting that the wells did monitor persistent gas influxes throughout the entire interval.**

Because the sub-permafrost accumulations are relatively shallow and under lower pressure, the gas will be much less dense, and thus voluminous, than conventional deeper accumulations. The exception to this is if the gas is in hydrate form where methane concentrations are some 160 times higher than in free gas form (Majorowicz and Hannigan, 2000). Table 6 shows the potential volumes of gas within the hydrate stability zone beneath Hopen using typical net to gross and reservoir properties for the De Geerdalen Formation (Mørk, 2013; Hynne, 2010). The calculations show volumes for both free gas, under atmospheric pressure and if it is in hydrate form.

Given the sparse data and bias in drilling locations it is impossible to be very quantitative with respect to the size and frequency of these accumulations. What is evident is that permafrost is acting as an ultimate seal to these accumulations, and that they are numerous, and, based on the only occurrence where flow was recorded, on the orders of million cubic metres.

**5.7 Regional distribution**

Based on the occurrences in Svalbard, the prerequisites for sub-permafrost gas to accumulate are, firstly, an impermeable (ice-saturated) permafrost layer, secondly, a source of gas and, finally, gas migration at a time after permafrost formation. Much of the Circum-Arctic shares a similar geological history with Svalbard. A major source of migrating gas in Svalbard is likely from the Mesozoic source rocks (Ohm et al., 2019), which can also be found in the Russian and North American Arctic (Leith et al., 1993; Polyakova, 2015). Recent uplift caused by isostatic rebound has left fluids in the subsurface on the Barents and Svalbard out of pressure equilibrium and driving present-day migration (Birchall et al., 2020). Svalbard shares its Pleistocene glacial history with the

Circum-Arctic (Batchelor et al., 2019) so it is not unreasonable to expect sub-permafrost gas accumulations to be regionally widespread. Indeed, gas emanating from zones of permafrost is well-documented onshore and offshore in the Russian Arctic, particularly in hydrocarbon provinces (Chuvilin et al., 2020 and references therein) and as natural gas hydrates (Yakushev & Chuvilin, 2000).

**6 Conclusion**

Although gas at the base of permafrost has been encountered frequently during more than fifty years of drilling in Svalbard, it has not been studied or widely recognised until now. In this study we have provided a synthesis of historical and modern observations and its implications.  With the Arctic warming faster than anywhere else on the planet (Lind et al., 2018), it seems likely that thawing permafrost in Svalbard will contribute to a positive feedback loop in releasing major amounts of trapped methane into the atmosphere. Furthermore, hydrates, which are probably present locally, are particularly susceptible to small changes in thawing permafrost and warming temperatures (Betlem et al., 2019; Sloan Jr et al., 2007). In a local context, it may be feasible to exploit these resources to provide a local source of power, which is currently reliant on coal.

The presence of permafrost associated gas and gas hydrates are an extremely important subject with recent studies showing the presence of shallow gas throughout the Circum-Arctic (Nielsen et al., 2014; Minshull et al.,

2020; Hodson et al., 2020; Chuvilin et al., 2020), much of which shares its geological history with Svalbard.

Insights into sub-permafrost dynamics from the vast number of economic boreholes through the permafrost in

Svalbard can be applied to analogous areas to understand and predict the ongoing processes.

[revised manuscript text omitted]

---

## Author Comment (AC1)

**Dear Reviewer**

**Firstly, we sincerely appreciate the time you have taken to review our article and found your comments very insightful and fair. We have answered your comments as best we can below.**

**I have included your original comments in black and written our responses in bold. We have also attached a pdf copy of this with our responses in red to better highlight the comments vs responses.**

**Yours sincerely, on behalf of the authors**

**Tom Birchall**

Comment 1: I found the overall presentation is not well structured and clear, and hence is difficult to review. This, in my option, is because the paper mostly focuses on the data description, and did not clearly show how they help to fill the knowledge gap. Based on the author guideline of TC, a review article should "summarize the status of knowledge". The paper in its current format is more like a datasets report rather than a scientific paper. This could be reflected by the Results section. It gives very detailed information on permafrost thickness, gas presence/absence, but lacks in-depth analyses.

**We agree very much with this, and we will aim to provide a clearer structure to the article. We appreciate that we need to better highlight the existing knowledge gaps and will add more detail of them at the end of the introduction – notable examples include the lack of previous publications even mentioning the trapped methane and the lack of existing wellbore data (direct or indirect) on the base permafrost and its depth, both of which are addressed by these vintage boreholes. We were requested to change this it to a review article by the editor. We see this article as a review of vintage well data (not published works). However, it is worth noting that with one exception, none of the wells that penetrate to base permafrost were drilled with the purpose of defining permafrost. Essentially, we have a great breadth of data over a very large area, but, as you mention, that leads to the lack of specific analyses. However, because this subject has not been previously addressed in Svalbard, and much of the data is in paper archives and not easily accessible to the scientific community, we feel it is important to provide as much information as possible.**

Authors used a large number of observations from various sources: hydrocarbon exploration, coal boreholes, and scientific boreholes. If I understand correctly, these wells and boreholes were instrumented in different periods, by different technicians as well as scientists, and for different purposes. In this case, the data is expected to have different degree of confidence. In section data and methods, authors should clearly clarify such degree of confidence, and the data quality control.

**This is a very good point, and we will clarify this. We found that the strength of our investigation is not in individual datasets of specific wells, but the amalgamation of broad datasets within a substantial region that our fairly diverse team has the expertise to interpret and integrate (often through ruling out alternative hypotheses rather) into the bigger overview.**

Last but not least, I would suggest authors double-check the definition of permafrost (please see Van Everdin-gen, R. O., 1998), the current definition in L50 is not true (please also see my specific comment).

). **permafrost**
    **Ground (soil or rock and included ice and organic material) that remains at or below 0°C for at least two consecutive years** (see Figure 2).
COMMENT:
    Permafrost is synonymous with perennially *cryotic ground*: it is defined on the basis of temperature. It is not necessarily frozen, because the freezing point of the included water may be depressed several degrees below 0°C; moisture in the form of water or ice may or may not be present. In other words, whereas all perennially *frozen ground* is permafrost, not all permafrost is perennially frozen. Permafrost should not be regarded as permanent, because natural or man-made changes in the climate or terrain may cause the temperature of the ground to rise above 0°C.
    Permafrost includes perennial *ground ice*, but not glacier ice or icings, or bodies of surface water with temperatures perennially below 0°C; it does include

**We appreciate the reference and paste the definition from the source for readers.**

**"Permafrost is defined as ground that remains at sub-zero (in Celsius) temperature for more than two consecutive years, regardless of fluid content."**

**I will rephrase to something more appropriate, e.g.:**

**"Permafrost is defined as ground that remains at or below zero degrees Celsius, temperature at least two consecutive years, regardless of whether the fluid is frozen." And will clarify subsequent sentence by what we mean with "ice-saturated permafrost".**

For these reasons, I would not recommend that the paper be published in TC in its current form. I would actually recommend authors significantly shorten the paper and focus on how the valuable data contribute towards knowledge of natural gas in/below permafrost. Another option would be thinking about publishing it in a data journal.

**We think this is a very useful comment and completely agree. We will will remove the majority of the hydrate and modelling side focus more on the trapped gas for a more concise manuscript with less uncertainty. One example that might be worth keeping is Adventdalen where we do have much stronger calibration points with respect to gas geochemistry and temperature data (that were collected by scientific boreholes with those datasets as a major goal).**

**Because Svalbard is representative of the high relief, rapidly uplifting, glaciated parts of the Arctic and that this is the first such investigation on the subject in Svalbard, we don't want to withhold data from the community. As a scientific community we need insights into this gas in the literature, not just because it is a hazard, but because of its important link with increasing glacial recharge (as the landscape approaches peak meltwater production) that is controlled by geological processes over much longer timescales (e.g., uplift, erosion, fracturing, abnormal fluid pressures) than we may often consider. These are systems that we cannot develop an understanding of from the more well studied areas such as Alaska, the McKenzie Delta or Siberian continental shelf environments.**

**I would also like to highlight that we have a team of Svalbardian experts from a variety of backgrounds working on this article, including permafrost, bedrock and petroleum and coal geology (and an employee from the local mining company), drilling, and fluid flow, which means we are in a unique position to be able to disseminate the broad datasets. This data is unique and will never be collected again as there is little appetite to drill expensive wellbores through the permafrost without economic incentive (scientific boreholes are common in the upper few metres,**

**but nowhere near reaching the sub-permafrost realm), hydrocarbon exploration ceased here long ago, and the last coal mine will shut down in Longyearbyen in 2023.**

P2, L41: Why relatively young permafrost is important here?

**We will rephrase that. The importance is that it indicates fluid migration is very recent – e.g., the past few kyr rather than 100s of kyr or myrs.**

P2, L50: … remains at or below 0°C for at least two consecutive…

**Addressed above.**

P2, L52: Physically speaking, ice-saturated permafrost possesses extremely good sealing, because of ? Also, this knowledge is similar to your conclusion 4: permafrost in valleys with more ice is a better seal. Then, why do you still need conclusion 4 if it has been widely known? P

**We will rephrase this – I should have explained it better that in terms of permeability it does, but there is some uncertainty about its extensiveness (e.g., can everything be draining through points such as pingos?) essentially it seems not. There is also the point that wellbores in the highlands seem to lack water/ice in the permafrost zone (so may be a worse seal for gas).**

3, L72--73: 100--500 m is quite thick… P7, L196: I agree identifying the permafrost thickness is challenging. On the other hand, permafrost is a hidden phenomenon, and identifying its presence is also NOT simple (even at a site scale) and may even be ambiguous without direct evidence (e.g., soil temperature and samples). Please see Cremonese et. al (2011).

**This is a good point; indeed, we rely on indirect indicators. We think adding more information, particularly from scientific wellbore DH8 (Longyearbyen CO2 Lab; Gilbert et al. 2018), which was drilled specifically to characterise permafrost, will provide insight here. We can provide details of ice content, rock properties and stratigraphy from this well. Although these wellbores do not penetrate the base permafrost, we do have a lot of expertise at UNIS in permafrost identification and characterisation. Though we do note that the base permafrost (or really, the base of impermeable ice) in our datasets does not have a single smoking gun indicator, rather an overwhelming number of cases where gas influxes into a well at a depth that does not correspond to an obvious geological break, and always occurs at an approximate place where we would expect the base permafrost to be (which is highly uncertain).**

---

## Author Comment (AC2)

Dear Associate Professor Knutsen,

Many thanks for taking the time to review our article and providing very useful and insightful comments.

I have included your comments in normal text below with our responses in bold. We have also attached a PDF version with our responses in red to help distinguish the comments vs responses.

With best regards
on behalf of the authors

Tom Birchall

The relationship between the topics of the paper is intriguing but also very complex. The current set-up of the paper, especially the methods and the discussion part, is a bit unfocused, and the many different topics appears to be treated a bit lightly.

**We agree and will restructure this to an extent. Addressing your next comment should also help to develop the focus of the article better.**

I would recommend the authors to focus on the observations of permafrost and maybe trapped gas, and how this is distributed in the Svalbard area. The distribution and timing of this might also be more elaborated. However, I am not sure if the link to petroleum system (or maybe release of gas to the atmosphere and climate implications) should be included. If the latter topics are to be included, the presentation and discussion of these should be strengthened.

**We think this is a good point, and we can tone down these aspects to make the paper clearer without removing the important data. Though we do believe it is important to be clear that gas is trapped, and that permafrost is very young clearly demonstrates migration is ongoing (both onshore and offshore). We will shorten the speculation with respect to the climatic impact and explain much more work is needed to quantify this. We also think removing some of the more speculative modelling of permafrost and hydrate stability zone will help bring more focus to the paper.**

Specific Comments in attached (tc-2021+226-RC2-supplement) supplement

**Your comments here are all very helpful and we will make the recommended amendments and elaborations. Some specific comments and responses are below:**

Line 69 – Please refer to figure 1 - and this figure might be bigger, maybe with some minor tectonic elements? (The Billefjorden Fault Zone?)

**We will do this. We will enlarge the map and include more tectonic elements (it is likely they are important for ongoing migration).**

Line 84: Generally: could it be possible to differentiate between the fjords and the west coast .. how far in the fjords are "inland"?

**Yes – we can bring in information from many of the good UNIS Arctic Geophysics papers on this (e.g. Skogseth et al., 2020). It also seems the thinking and evidence around coastal permafrost are changing (since writing this manuscript even) so we will update this.**

Line 102 Suggest to rephrase this section - or remove. Generally the manuscript could be strengthened by deciding how much "operational" parts should be included: these parts are not the always the best parts..

**We will remove this.**

Line 267 Locations in Figure 1? Could it be possible to include depths to base permafrost and shows in the table?

**Yes – a good idea and we will do this in the revision.**

Line 308 Generally the "model and parameters" might be better presented and explained.

**We agree and actually think that removing much of these models will benefit the paper. Based on responses we feel the manuscript would benefit from being based on more certain observations than the more speculative modelling. The case of Adventdalen we feel may still be of use where we have calibration points, (which we will be careful to provide more details of.**

A general comment: the language is in places very "oral" - that might be a the authors personal approach, but can also in places overshadow the message..

**Noted and we will amend this.**

The overall discussion of pressure / aquifers and thin/cold based glaciers is important - and complex. Please consider to elaborate (or leave it "out").

**A good point – we think this is probably worth its own paper at some point in the future.**

Line 590 - A map showing the areas with permafrost (observed / modelled) would be very helpful!

**Agreed and we will do this.**

---

## Author Comment (AC3)

**Dear Dr. Dallimore**

**We sincerely appreciate the considerable time you have taken to review our manuscript. We are also grateful for the comments in the manuscript attachment which will undoubtedly improve the manuscript. While we cannot fix the data we have, I hope we can provide some answers and comments below. Your comments are in normal text and our responses are in bold. We have also attached a version with our responses in red to help distinguish comments from replies.**

**Many thanks and best regards**

**On behalf of the authors**

**Tom Birchall**

**Please note that our overarching response is at the end of this document. However, we have addressed some of the more specific comments below.**

**Comments from manuscript attachment**

**The comments within the supplement are extremely useful and we will make those changes as they will improve the manuscript. The only specific comment we address here is:**

P 15 - There is no convincing evidence presented indicating presence of pf in nearly all these well presentations.

**We completely agree – perhaps we were unclear with this but it was more to show that the petrophysics here is extremely challenged (or simply impossible)in identifying permafrost directly. The more important thing is to note the lithology (these wells were cored) and where gas influxes happened (e.g. in the middle of a sandstone with no other explanation for a top seal – including cementation etc. Cores were generally more useful than petrophysical data).**

Unfortunately, as a permafrost scientist I was disappointed that the available data presented in the paper to demonstrate permafrost occurrence, ice bonding and associated permeability reduction is in my view rather weak and not adequate for publicatiion as submitted. I have studied well log interpretations of permafrost occurrence in hydrocarbon wells in the North Slope of Alaska, the Mackenzie Delta and the Beaufort Sea. As described in the paper there are many published studies from these settings using industry data to delineate well-defined occurrences of permafrost based using mainly well log resistivity and seismic velocity anomalies as well as in situ temperature measurements. Many of these wells have additional well log data and core samples that reveal that the permafrost occurs in mainly unconsolidated high porosity sediments, rather than in cemented bedrock. Based on the presentations in the Birchall et al paper it seems that the shallow sections of the hydrocarbon wells and scientific wells from Svalbard were not fully characterized by industry with the same rigor that we see in some other settings. This deficit in hard data is a challenge for the authors to overcome if they wish to present convincing and strong arguments in support of their conclusions. Indeed in the various well log sections presented in the paper, I did not see any

examples that in my opinion yielded a high confidence assessment of ice bonded permafrost occurrence with multiple lines of evidence such as documented in other settings. Without well log indicators, the authors rely heavily on assumed variations in permeability expected from ice bonded permafrost to non-ice bonded sediments resulting in water influx into well bores or simply the occurrence of free gas.

**We will provide information from the DH8 wellbore from the Longyearbyen CO2 Lab that was drilled and fully cored to characterise permafrost (but did not penetrate its base). We have provided temperature where it is available but see below for a more clear example from Beka et al. (2017). Note that we are very aware that we cannot directly see permafrost evidence in the petrophysical data (we have three experienced petrophysicsists who have tried), we elaborate on that later in this document.**

 *T.I. Beka et al. / Journal of Applied Geophysics 136 (2017) 417–430*

[Figure]

**Fig. 8.** Resistivity logs of Dh1 (a), Dh2 (b), Dh4 (c) boreholes (see Fig. 1b for locations) compared with the final 2D resistivity model averaged between the sites T08–s09 and presented as a 1D curve in (c). In addition, temperature log and lithology are displayed for each borehole. As the TEM and MT result, the shallow resistivity log from Dh1 and Dh2 indicate lower resistivity for the upper ca. 50 m depth and suggest a decreasing tendency below ca. 350 m in a similar way as the modelled resistivity result towards the décollement zone (the black section of Dh4's stratigraphy). For more details on the wireline logging refer to Elvebakk (2010).

This association in itself is not an unreasonable expectation, however the authors have not given the reader the basic information they need to appraise these indicators. While there are some vague statements suggesting that the study wells penetrate low porosity bedrock rather than unconsolidated sediments, no data is presented on the petrophysics of bedrock occurrence in terms of porosity and permeability and only a few examples of pore water geochemistry.

**We agree on this – we can provide more information on this from CO2 Lab data and publications (see below for an example of figure – made by K. Senger, one of the authors here). and we can add information on the poro-perm and fractures from Adventdalen Note that most data from the economic wellbores throughout Svalbard were not drilled by permafrost scientists, thus shallow pore water geochemistry data is not collected (however it is something that has been studied at pingo sites (e.g. Hodson et al., 2019 & 2020).**

[Figure]

All of these factors are critical for the assessment of the manifestation of ice within the porosity of the bedrock setting or occurrence of ice in fractures limiting fracture permeability. I have not rejected the paper as it is my hope that the authors in a revision of the paper can provide an expanded assessment of the petrophysics of the setting they are studying. Hopefully bedrock mappers or industry scientists have appraised the porosity, permeability and fracture characteristics of bedrock occurrences. If these observations are available they should be described for the reader and interpreted within the context of the goals of the paper.

In addition, it is important to understand if there are any observations of ground ice form and occurrence that have been complied by surface mapping or in laboratory studies on core samples. Another concern is the confidence in assessment of the base of permafrost. No in situ ground temperature data are presented on permafrost occurrence and for the most part the authors have had to rely mainly on permafrost modelling by others or very weak estimates based on drill mud temperatures. I note that on Figure 3 for instance that the authors acknowledge considerable uncertainty in their estimates of base of permafrost.

**We agree, the base permafrost depths are uncertain. However, we are more concerned with what is happening below the permafrost and not characterising it. This is very true, the base permafrost**

**estimates are highly uncertain and we address this later. We have also noticed that temperature data have become obscured in Figure 5 and needs to be fixed.**

Gas and water anomalies in many wells are considerable distance below their high confidence estimates (see my notes on the pdf copy of the manuscript). This does not lend confidence to the assertions made in the paper as it may be that the indicators are not related to variation in ice bonding. Concerns related to the geothermal setting, pore fluid chemistry and petrophysics also in my view render the gas hydrate discussion in the paper as even more speculative than the permafrost discussion. Indeed in several places in the text the authors suggest that if gas occurred in hydrate form the volume of stored gas could be much larger than if the gas was free gas. This suggestion is in my view unsupported, leading to my suggestion that reference to gas hydrate occurrence in the paper be significantly scaled back.

**We very much agree, and the permafrost and hydrate modelling parts will be significantly scaled back and only kept where we have reasonable calibration points in Adventdalen.**

I close with a note of encouragement to the authors - the topic they are studying is important and certianly of interest to the readers of the journall, thus I hope that they can continue their study and advance this paper to publication. I have made some minor comments on the attached PDF that I hope will help point out to the authors my concerns with the present manuscript. These include a need for considerable effort to improve nearly all of the figures and the captions for the figures and tables. I have not dwelled on the writing however there is also a need to stream line the writing (perhaps by simplifying some of the extended discusison on the petroleum setting) and improve the consistency in terminologies referring to permafrost terminology.

**We will cut back on the petroleum systems part and improve the figures as suggested.**

Finally, I recommend that the authors more carefully assess references from the Mackenzie Delta. This could be done simply by using Googlescholar with a search for 'permafrost occurrence Mackenzie Delta'. In particular perhaps the Geochemistry paper by Collett and Dallimore and Geology paper by Dallimore and Collett would be helpful as they provide some insights and observations of gas within and below ice bonded permafrost

**Perhaps we were unclear in the aim of our manuscript and this is something we will rectify. The focus of this paper was intended to be to highlight gas accumulations and fluids below the permafrost in Svalbard, something that has not been documented before.**

**Perhaps we were not clear enough in the aims, and why it is important we utilise this vintage data - the petroleum wells were drilled between the 1960s and 1990s and coal boreholes sometimes predate that. Very few recent wellbores penetrate the entire permafrost interval, and even fewer are likely to do so in future due to their high cost and the cessation of economic drilling in the archipelago. Although they were not focussed on characterising the base permafrost, they made many important observations, most notably large pockets of shallow gas (even blowouts) that could not be explained by conventional trapping methods. Because these data have not been previously looked and the topic of permafrost trapped gas has not been studied in Svalbard, which**

is warming faster than anywhere in the Arctic. We feel it is important to disseminate it to the scientific community as a team of authors with a variety of geoscientific backgrounds.

While we agree that the details in the data are not heavily focussed on permafrost, it is to be expected when the vast majority of boreholes that penetrate to the base of permafrost were in no way targeting it. Only petroleum wellbores have collected petrophysical data, and rarely a full suite of logs over the shallow intervals. The coal exploration wellbores do not collect any petrophysical data, but do fully core them and record depths where gas influxes occur.

We should also be very clear that Svalbard's geology and geological history is very different to that of the McKenzie Delta or North Slope of Alaska. In Svalbard the base permafrost is invariably in bedrock that has previously been buried to several kilometres depth and subsequently uplifted. Therefore, the already highly lithified, low porosity and heterogenous rocks show very little petrophysical contrast to fluid type or phase changes, though still provide good indicators to lithology. It is not the same as ice-bonded hard vs unconsolidated sediment that we see in other parts of the world (and also provides challenges for geophysical methods). I can assure you we did look closely at the petrophysical data (we actually used your very nice McKenzie work to see if it could be applied here) – three of our authors have worked extensively with petrophysical data in our careers (one has just returned as a petrophysicist on the latest IODP campaign). Ultimately, we found that petrophysical data is only useful at determining lithology, not at identifying permafrost (but we can make a figure to demonstrate this if it is useful).

It is because of the differences in geomorphology and geological histories between Svalbard and the well-studied parts of North America that make providing these data and observations so important to the scientific community. Svalbard is representative of the high relief, rapidly uplifting, glaciated parts of the Arctic and this is the first such investigation on the subject of sub-permafrost trapped methane in Svalbard. As a scientific community we need insights into this gas in the literature, not just because it is a hazard, but because of its important link with better understanding glacial recharge and the geological controls that occur over longer timescales than glaciologists often consider (e.g. uplift, erosion, fracturing, abnormal fluid pressures). These are systems that we cannot develop an understanding of from areas with better and recent datasets such as the McKenzie Delta, Alaska's north slope, Siberian continental shelf environments.

By far the biggest strength of this drilling-focused dataset is in its diversity and sheer number of wells that have encountered shallow gas. When combining this with well reports, gas influx data (even blowouts), and the geology, we can eliminate alternative hypotheses (e.g., lithological seals). So, while there certainly is no individual piece of data that provides a smoking gun, the evidence is there, and we should probably be clearer in describing that.